# Explaining Data Mixing Scaling Laws

**Rui Dai**[1]  **Shuran Zheng**[2]

## Abstract

Recent research has established empirical scaling laws to predict model performance on multi-domain data mixtures. However, a theoretical understanding of these model loss behaviors remains absent. In this work, we propose a unified framework to explain the underlying mechanics of data mixing. Our approach extends theoretical perspectives originally developed for standard neural scaling laws (e.g., Kaplan and Chinchilla) to the multi-domain setting. Based on the distributional assumption that domains overlap on fundamental skills while diverging on specialized skills, we identify two key factors that govern the domain losses of models trained on different data mixtures: *Capacity Competition*, where the allocation of finite model capacity couples domain losses globally, and *Noise Reduction*, where optimal weights shift toward harder-to-learn domains to minimize overall noise. Empirical evaluations show that our framework outperforms existing baselines by fitting the loss landscape with a lower Mean Relative Error and identifying higher-performing training mixtures. Most importantly, our model successfully extrapolates across scales, predicting highly effective mixtures for large, unseen scales using parameters fitted on smaller ones. In addition, our model achieves these results using significantly fewer parameters compared to previous empirical laws.

## 1. Introduction

Large foundation models are typically trained on data from multiple domains, with the data mixture—the proportion of each domain used—playing a critical role in model performance. However, discovering the optimal data mixture is often a highly costly process that lacks principled methodologies. Practitioners often rely on expensive trial-and-error or static heuristics.

To address this, recent research has moved toward principled methods that generally fall into two distinct paradigms: online adaptation and offline prediction. Online methods attempt to adjust domain weights dynamically during the training process based on the model's ongoing learning trajectory (Albalak et al., 2023; Chen et al., 2023; Jiang et al., 2025; Chen et al., 2025; Li et al., 2026b). While these approaches can be effective in dynamic settings, they often incur computational overhead and remain theoretically opaque regarding how dynamic weighting ultimately impacts model's capabilities.

In parallel, a prominent line of offline research has focused on predicting the loss landscape *a priori* through data mixing scaling laws (Shukor et al., 2026; Ye et al., 2025; Ge et al., 2025; Kang et al., 2025) and determining the mixture before training begins. These empirical frameworks attempt to predict a model's test loss on specific domains as a function of the mixing weights used during training. Several functional forms have been proposed to model this relationship, as summarized in Table 1. These laws deviate from the standard power-form scaling laws and present non-trivial domain interaction: the loss on a domain depends not only on the weight of the domain itself, but also the weights of other domains, and this correlation does not exhibit a simple functional form.

*Table 1.* Examples of empirical functional forms for predicting domain loss $L_i(h)$ based on mixture weights $h$. $N$ and $D$ represent model parameters and training tokens, respectively.

| Reference | Functional Form $(f_i(h, N, D))$ |
|---|---|
| (Shukor et al., 2026) | $L_i \approx E_i + \left( \sum_{j=1}^{K} C_{ij} h_j^{\gamma_{ij}} \right)^{-1}$ |
| (Ye et al., 2025) | $L_i \approx c_i + k_i \exp\left( \sum_{j=1}^{K} t_{ij} h_j \right)$ |

Despite the practical utility of both dynamic algorithms and offline empirical fits, a rigorous theoretical understanding of the mechanics driving domain interaction remains largely absent. Focusing specifically on offline mixing laws, relying

[1]Beijing Institute of Technology [2]IIIS, Tsinghua University, Beijing, China. Correspondence to: Shuran Zheng <shuranzheng@mail.tsinghua.edu.cn>.

*Proceedings of the 43^{rd} International Conference on Machine Learning*, Seoul, South Korea. PMLR 306, 2026. Copyright 2026 by the author(s).

solely on empirically fitted curves presents a significant bottleneck. Not only is the fitting process resource-intensive, but the resulting laws act as "black boxes": it is unclear whether they generalize to larger scales or different datasets, nor is it obvious how to map the predicted domain loss to downstream task performance.

In this work, we propose a unified theoretical framework to explain the underlying mechanics of data mixing. Our framework extends previous theoretical frameworks for standard neural scaling laws—specifically the Quantization Model (Michaud et al., 2023; Liu et al., 2025b) and the Projected Linear Regression Model (Lin et al., 2024a; Bordelon et al., 2024)—to the multi-domain setting. Based on a natural distributional assumption that different domains overlap on fundamental skills and diverge on specialized skill, we identify two key factors that decide the loss of models trained on different data mixtures:

- **Model Capacity Competition:** The model has a finite capacity and can only learn a finite number of skills. The specialized skills from different domains compete for the model capacity. Adjusting domain weights will change the importance of the skills and thus change the model capacity allocated to each domain. The resulting model capacity allocation introduces a non-trivial domain interaction, and is a key factor that determines the trained model's loss on a domain.

- **Noise Driven by Data Amount:** For each skill within the model's capacity, the loss incurred by the skill depends on the number of times that the model has seen the skill. As skills from different domains have different difficulty levels, the loss decreases at different speed as the domain weight increases; this dynamic shifts the optimal mixture weights toward domains that are harder to learn.

Leveraging this theoretical framework, we formulate loss prediction as a convex program that yields numerical estimates for arbitrary mixtures. Furthermore, we frame the search for the optimal training mixture as a bi-level optimization problem, which can be efficiently solved using Online Mirror Descent.

Empirically, our results validate the theoretical framework across several key dimensions:

- **Superior Fitting Accuracy:** Our models fit the observed loss landscape with a lower Mean Relative Error (MRE) than existing empirical scaling laws.

- **Optimal Mixture Prediction:** Our framework effectively identifies optimal training mixtures that yield the lowest test loss on the target average distribution.

- **Cross-Scale Extrapolation:** Most importantly, our framework successfully extrapolates across scales, pre-

dicting highly effective mixtures for large, unseen model and dataset sizes using parameters fitted exclusively on smaller ones.

- **Parameter Efficiency:** Crucially, we achieve these results while utilizing significantly fewer free parameters compared to leading empirical laws.

## 2. Related Work

**Data Mixture Selection.** Optimizing the pre-training data mixture is critical for maximizing downstream model performance. Data mixture can operate at various granularities, ranging from fine-grained token-level selection (Lin et al., 2024b) to coarser domain-level mixture. Domain-level approaches are particularly advantageous due to their superior computational efficiency. While early domain-level data mixture approaches relied on static heuristics or expensive trial-and-error, recent research has gravitated towards principled, compute-efficient strategies. These approaches generally fall into two paradigms: offline selection prior to training and online adaptation during training.

**Offline: Data Mixing Laws.** This line of work establishes empirical scaling laws for data mixtures to predict the loss landscape as a function of mixture ratios, from which the optimal mixture is derived. Ye et al. (2025) proposed an exponential-form data mixing law, which is extrapolated to larger model and data scales via the standard power-law scaling. Ge et al. (2025) and Kang et al. (2025) introduced laws that jointly consider mixture ratios and data size (or training steps). Most recently, Shukor et al. (2026) formulated a unified law that explicitly incorporates model size, dataset size, and domain mixture ratios into a single scaling law. Other studies address specialized settings, including high-quality domain data (Gu et al., 2025), and data-constrained scenarios involving repeated tokens (Muennighoff et al., 2023).

**Offline: Proxy-Based Selection.** These methods determine optimal domain weights on small proxy models and transfer them to larger target models (Xie et al., 2023; Fan et al., 2023; Liu et al., 2025a; Zhang et al., 2025; Diao et al., 2026; Wettig et al., 2025). Recently, Magnusson et al. (2025) investigated the efficacy of proxy mixtures when scaling to large models.

**Online Approaches:** In contrast to static offline selection, online methods adjust domain weights dynamically during the training process, including ODM (Albalak et al., 2023), Skill-it (Chen et al., 2023), Aioli (Chen et al., 2025), ADO (Jiang et al., 2025), and PiKE (Li et al., 2026b). While these approaches can be effective, they often incur computational overhead and require additional validation loops during training.

**Theoretical Foundations of Neural Scaling.** While empirical research has established that neural network loss scales as a power law with respect to model size, data size, and compute (Kaplan et al., 2020; Hoffmann et al., 2022), deriving these exponents from first principles remains a central challenge. Two primary lines of research seek to explain the underlying mechanics of neural scaling laws:

- **Linear Model Analysis:** One stream of research has sought to derive scaling laws through the analysis of linear models. Initial efforts focused on simplified environments, such as regression on fixed-dimension manifolds (Sharma, 2022). Maloney et al. (2022) and Bahri et al. (2024) expanded this setting to include generative data models and random feature models, demonstrating that power-law scaling arises in the dual limit of infinite data and infinite parameters. A more recent body of work focused on the training dynamics, specifically tracking one-pass Stochastic Gradient Descent within linear frameworks (Fonseca et al., 2024; Atanasov et al., 2024; Lin et al., 2024a; Bordelon et al., 2024; 2025; Li et al., 2026a). Bordelon et al. (2024) applied dynamic mean field theory to randomly projected linear models, recovering power laws in the asymptotic limit. Lin et al. (2024a) utilized a similar randomly projected linear model to reconcile neural scaling with traditional statistical theory, offering an explanation for why the classical variance error is unobservable when fitting the neural scaling law empirically.

- **Skill Learning:** Another line of work abstracts from complex training dynamics, instead viewing scaling laws as a consequence of "skill learning" (Michaud et al., 2023; Liu et al., 2025b; Arora & Goyal, 2023; Pan et al., 2026). Michaud et al. (2023) proposed the Quantization Model, which frames learning as the sequential acquisition of discrete "quanta" of skills distributed according to a power law, thereby recovering the observed power-form scaling. This framework was recently extended by Liu et al. (2025b) to three models with different levels of complexity.

# 3. Preliminaries and Problem Description

This section establishes the theoretical background for single-domain neural scaling laws and formally defines the multi-domain data mixing problem.

## 3.1. Theoretical Foundations of Single-Domain Scaling

Neural scaling laws describe a predictable power-law relationship between test loss $L$, model size $N$, and dataset size $D$ (Kaplan et al., 2020; Hoffmann et al., 2022):

$$L(N, D) \approx \frac{A}{N^\alpha} + \frac{B}{D^\beta} + E.$$

Below, we review two major frameworks that attribute this power-law scaling to the intrinsic power-law structure of the data distribution.

**Quantization Model.** Michaud et al. (2023) posit that knowledge within training corpora decomposes into discrete skills, or "quanta," which follow a Zipfian distribution: $p(q_k) \propto k^{-\alpha}$ for $\alpha > 1$. When training a model of size $N$, it learns the top $N$ most frequent quanta to minimize expected loss. Assuming each unlearned quantum contributes a constant error $c$, the loss is governed by the unlearned tail:

$$L(N) = c \sum_{k=N+1}^{\infty} p(q_k) \approx c \int_N^\infty (\alpha-1)k^{-\alpha}\, dk = c \cdot N^{-(\alpha-1)}.$$

**The Linear Regression Model.** To account for the stochasticity of training dynamics, Bordelon et al. (2024); Lin et al. (2024a) analyze linear regression under one-pass SGD. They assume data covariates $\mathbf{x} \in \mathbb{R}^d$ have a covariance matrix $\mathbf{H}$ with power-law decaying eigenvalues $\lambda_k \propto k^{-\alpha}$. To model a neural network with finite capacity $N$, the input is projected via a "sketching matrix" $\mathbf{S} \in \mathbb{R}^{N \times d}$. Minimizing the squared error of this projected model yields a scaling law capturing both capacity and data limits:

$$L(N, D) \approx \underbrace{O\left(\frac{1}{N^{a_1}}\right)}_{\text{Model Scaling}} + \underbrace{O\left(\frac{1}{D^{a_2}}\right)}_{\text{Data Scaling}} + E.$$

We defer the details of this projected linear regression model to Section D.1.

## 3.2. Empirical Data Mixing Laws

Consider a mixture of $K$ data domains $\mathcal{D} = \{\mathcal{D}_1, \ldots, \mathcal{D}_K\}$ sampled according to weights $h \in \Delta^{K-1}$. The primary objective of data mixing is to find the target-aligned training weights $h^*$ that minimize the model's loss on a specific target distribution defined by importance weights $w \in \Delta^{K-1}$, under a fixed compute budget parameterized by model size $N$ and training token count $D$.

Because exhaustively training large models to find $h^*$ is computationally prohibitive, practitioners fit empirical scaling laws to predict the held-out test loss $L_i$ for each domain: $L_i(h, N, D) \approx f_i(h, N, D)$. Once $f_i$ is fitted using small-scale proxy models, the optimal mixture $h^*$ is estimated by solving:

$$h^* = \underset{h \in \Delta^{K-1}}{\arg\min} \sum_{i=1}^{K} w_i f_i(h, N, D).$$

Table 3 summarizes key functional forms established in the literature.

# 4. Theoretical Framework for Data Mixing

In this work, we propose a unified theoretical framework to explain the underlying mechanics of data mixing. We extend two single-domain perspectives—the Quantization Model (Michaud et al., 2023) and the Linear Regression Model (Lin et al., 2024a)—to the multi-domain setting. We first introduce the Extended Quantization Model, which frames training as a capacity allocation problem. This serves as the foundation for the Extended Linear Regression Model, which implicitly solves this allocation problem while incorporating data-dependent noise.

## 4.1. The "Shared Head, Disjoint Tail" Structure

Building on (Pan et al., 2026), we introduce a natural structural assumption regarding how information overlaps across different data domains (e.g., Code, Math, English):

- **Power-Law Distribution:** Within each domain $i$, knowledge units (skills) follow a power-law distribution in terms of frequency.

- **Shared Head:** Different domains largely overlap in the head of the distribution—the region of high-probability, fundamental skills (e.g., basic grammar, logic, or arithmetic).

- **Disjoint Tail:** As we move to the tail of the distribution (rare, specialized knowledge), the domains become increasingly distinct and independent.

While real-world data is unlikely to be strictly disjoint in the tail, this idealization serves as a useful approximation. Based on our modeling in the following sections, when the data size is fixed and only the mixture weights vary, the change in loss induced by overlapping skills is likely to be small compared to the change induced by disjoint skills. In other words, the loss induced by overlapping skills will behave more like a constant compared to disjoint skills. We discuss the robustness of this approximation in detail in subsequent sections. Furthermore, we empirically validate our model against violations of this assumption through a synthetic stress test (Figure 1), demonstrating that predictive accuracy remains stable even as tail overlap increases.

## 4.2. Extended Quantization Model

We first extend the Quantization Model (Section 3.1) to multiple domains.

**Multi-Domain Capacity Allocation.** We associate domain $\mathcal{D}_i$ with a continuous skill space $k_i \in [1, \infty)$ having power-law density $p_i(k_i) = b_i k_i^{-(b_i+1)}$, where $b_i = \alpha_i - 1 > 0$. Under our structural assumption, all domains share fundamental skills $k_i \in [1, H]$. Assuming a model

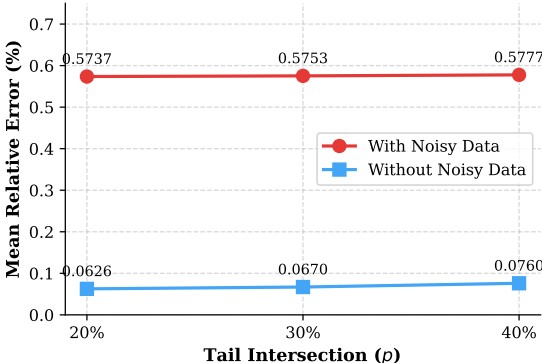

*Figure 1.* Fitting errors of our disjoint-tail theoretical model on synthetic data under varying degrees of actual tail overlap ($p$). We evaluate the model under two experimental settings: a noiseless setting and a noisy setting where Gaussian noises drawn from $\mathcal{N}(0, 1)$ are added to the losses. The plot demonstrates that the Mean Relative Error (MRE) remains consistently low and stable even as tail intersection increases up to $40\%$, highlighting the robustness of our proposed model.

with size $N$ learns all shared skills, training reduces to allocating the remaining capacity to disjoint tails. For a training mixture $h \in \Delta^{K-1}$, the model selects coverage thresholds $x_i \geq H$ for each domain $i$ to minimize expected training loss:

$$
\begin{aligned}
\min_x \quad & \sum_{i=1}^{K} h_i c_i x_i^{-b_i} \\
\text{s.t.} \quad & \sum_{i=1}^{K} (x_i - H) \leq N - H, \quad x_i \geq H, \ \forall i.
\end{aligned}
\tag{1}
$$

Here, $c_i$ is the error per unlearned quantum. The optimal allocation $x^*(h)$ yields expected domain test losses $L_i(h) = c_i(x_i^*(h))^{-b_i} + E_i$ with an additional irreducible loss $E_i$.

**Domain Interaction.** The capacity constraint forces domains to compete for resources. Using Lagrange multipliers (assuming large $N$ and similar $b_i \approx \bar{b}$), the optimal threshold $x_i^*(h)$ scales inversely with the "aggregate demand" of all domains: $\sum_k (b_k c_k h_k)^{\frac{1}{b_k+1}}$. Thus, $L_i(h)$ is tightly coupled to the weights and complexities of all competing domains.

**Tail Overlap.** Now suppose the tails of different domains overlap. A shared skill's aggregate probability density $p(\text{skill})$ will be relatively stable as the mixture shifts, making its status (learned vs. unlearned) less sensitive to weight changes. So the loss induced by overlapping skills will behave more like a constant compared to disjoint skills.

**Limitation.** A critical limitation arises when predicting the optimal mixture. Consider solving the bi-level problem to find the training mixture $h^*$ that minimizes the expected

test loss on a target distribution $w$:

$$h^* = \arg\min_h \sum_{i=1}^{K} w_i L_i(h) = \sum_{i=1}^{K} w_i \left( c_i (x_i^*(h))^{-b_i} + E_i \right).$$

(2)

Here $\sum_i w_i E_i$ is a constant and can thus be removed. As a result, the symmetry between the inner (Eq. 1) and outer (Eq. 2) objectives trivially yields $h^* \equiv w$. This contradicts empirical observations, where optimal training mixtures deviate significantly from the target. To resolve this, we introduce the Extended Linear Regression Model.

### 4.3. Extended Linear Regression Model

To address the limitations of the Extended Quantization Model and incorporate training dynamics, we extend the linear regression framework of (Lin et al., 2024a) to the multi-domain setting. This Extended Linear Regression Model can be viewed as an extension of the previous Extended Quantization Model as well: the training process implicitly solves the capacity allocation problem defined in (1), while introducing an additional noise term.

**Problem Formulation.** Following Section 3.1, we consider a linear regression problem over a union of $K$ domains. For each domain $i \in \{1, \ldots, K\}$, input covariates $\mathbf{x}_i \in \mathbb{R}^d$ (where $d$ can be infinite) are drawn from a distribution with zero mean and a covariance matrix defined as $\mathbf{H}_i = \mathbb{E}_{\mathcal{P}_i}[\mathbf{x}_i \mathbf{x}_i^\top]$. The label $y$ is generated by a global linear teacher $y_i = \langle \theta^*, \mathbf{x}_i \rangle + \epsilon_i$, where $\theta^* \sim \mathcal{N}(0, \mathbf{I})$ is the ground-truth parameter and noise $\epsilon_i \sim \mathcal{N}(0, \sigma_i^2)$. We consider a mixture distribution $\mathcal{P}(h) = \sum_{i \in [K]} h_i \mathcal{P}_i$ defined by weights $h \in \Delta^{K-1}$ with covariance matrix $\mathbf{H}(h) = \sum_{i=1}^{K} h_i \mathbf{H}_i$. Following (Li et al., 2026a), we model a neural network with $N$ parameters by projecting the high-dimensional input $\mathbf{x}$ into a $N$-dimensional feature space using a "Top-N sketching matrix" $\mathbf{S} \in \mathbb{R}^{N \times d}$ (detailed in Section D.2). The model learns a weight vector $\theta \in \mathbb{R}^N$ by minimizing the squared error on the projected features $\widetilde{\mathbf{x}} = \mathbf{S}\mathbf{x}$.

**Spectral Assumption: Shared Head and Disjoint Tails.** We then formalize the "Shared Head, Disjoint Tail" assumption. In spectral analysis, an eigenvector represents a pattern of variation in the data (e.g., a specific texture in images or topic in text), while its corresponding eigenvalue quantifies the variance (or strength) of that pattern. Intuitively, we assume real-world data consists of *universal patterns* shared across all domains (the head) and *specialized nuances* unique to each domain (the tails). To make the analysis tractable, we assume that all covariance matrices $\{\mathbf{H}_i\}_{i=1}^{K}$ share a common orthonormal basis of eigenvectors $\mathbf{U} = [u_1, \ldots, u_d]$ (i.e., they are simultaneously diagonalizable). In this basis, each matrix $\mathbf{H}_i$ is diagonal with

eigenvalues denoted by $\lambda_k^{(i)}$. We classify these eigenvectors into two categories:

- **Shared Head ($k \leq H$):** The first $H$ eigenvectors $\{u_1, \ldots, u_H\}$ represent universal components. We assume all domains possess non-zero variance along these directions ($\lambda_k^{(i)} > 0$ for all $i$), meaning these patterns are present in every domain.

- **Disjoint Tail ($k > H$):** The remaining eigenvectors represent domain-specific components. We assume each domain $i$ possesses a unique set of eigenvectors $\{u_k^{(i)}\}$ that are orthogonal to the specific components of other domains. Following Section 3.1, we assume that the variance along these directions follows a domain-specific power law:

$$u_k^{(i)\top} \mathbf{H}_j u_k^{(i)} = \begin{cases} k^{-\alpha_i} & \text{if } j = i \\ 0 & \text{if } j \neq i \end{cases}.$$

Consequently, for $k > H$, the spectral structures are completely decoupled: each domain $i$ has positive eigenvalues $\lambda_k^{(i)} = k^{-\alpha_i}$ along its own unique eigenvectors and zero along the eigenvectors of others.

Under this assumption, the mixture covariance matrix $\mathbf{H}(h) = \sum h_j \mathbf{H}_j$ exhibits a decoupled structure in the tail. Because the domain-specific eigenvectors do not overlap, the eigenvalue of a unique component in the mixture is simply its original variance scaled by the domain's proportion $h_j$. Specifically, for a tail eigenvector $u_k^{(i)}$ belonging to domain $i$, the corresponding eigenvalue in the mixture is:

$$\lambda(\mathbf{H}(h), u_k^{(i)}) = \sum_{j=1}^{K} h_j \cdot u_k^{(i)\top} \mathbf{H}_j u_k^{(i)} = h_i k^{-\alpha_i}.$$

**Connection with the Extended Quantization Model.** Here, each eigenvector $u_k^{(i)}$ (a pattern of variation in the data) can be viewed as a skill to be learned, and its corresponding eigenvalue $\lambda_k^{(i)} = h_i k^{-\alpha_i}$ (the variance of that pattern) can be viewed as the expected loss incurred if this skill is not learned (analogous to the $c(\text{skill}) \cdot p(\text{skill})$ term in our previous discussion). Therefore, our spectral assumption parallels the skill distribution assumption in the Extended Quantization Model. Furthermore, adjusting a domain's weight $h_i$ produces an equivalent effect in both frameworks: reducing the weight $h_i$ linearly shrinks all eigenvalues of domain $i$ according to $\lambda_k^{(i)} = h_i k^{-\alpha_i}$; similarly, in the Extended Quantization Model, reducing $h_i$ linearly shrinks the expected loss of skills in domain $i$ by shrinking $p(\text{skill})$.

**Loss Analysis.** Under the spectral assumptions established above, we analyze the expected test loss $L(h, N, D)$

of a projected linear model $\theta \in \mathbb{R}^N$ trained via one-pass SGD on a dataset of size $D$, sampled from a mixture distribution with weights $h \in \Delta^{K-1}$. The expected loss is governed by two primary factors. First, the training process implicitly solves the capacity allocation problem defined in the Extended Quantization Model (1), distributing the capacity $N$ across domains to the most valuable skills; consequently, the unlearned skills in each domain contribute a loss similar to $L_i(h) = c_i(x_i^*(h))^{-b_i}$ in the Extended Quantization Model. Second, for each skill within the model's capacity, the loss will depend on the number of times it has been observed; in other words, the stochastic nature of one-pass SGD introduces a noise term to the domain test loss $L_i$, which is determined by the number of training samples drawn from that domain.

**Theorem 4.1** (Informal). *Given data size $D$ and model size $N$, assume domains have mutually disjoint tails with eigenvalues $\propto k^{-\alpha_i}$ ($\alpha_i > 1$) and negligible shared head error. Given a training mixture $h \in \Delta^{K-1}$, let $b_i = \alpha_i - 1$, $c_i = 1/(\alpha_i - 1)$, and let $x^*(h, N)$ be the optimal solution of the Extended Quantization Model (1) defined by model capacity $N$, mixture weights $h$, parameters $\{b_i\}$ and $\{c_i\}$. For a projected linear model $\theta \in \mathbb{R}^N$ trained via one-pass SGD on $D$ samples drawn from a mixture $h$, its expected test loss on domain $i$, denoted $L_i(h, N, D)$, satisfies*

$$L_i(h, N, D) \approx c_i x_i^*(h, N)^{-b_i} + A_i(Dh_i)^{-a_i} + E_i \quad (3)$$

*where $a_i, A_i, E_i$ are constants that depend on $\alpha_i$.*

We defer the formal theorem to Section D.8.

**Optimal Mixture and Symmetry Breaking.** While capacity competition $x_i^*(h, N)^{-b_i}$ drives domain interaction, the noise term $A_i(Dh_i)^{-a_i}$ depends solely on $h_i$. When finding the optimal training mixture $h^*$ for a target $w$:

$$h^* = \arg\min_h \sum_{i=1}^K w_i\big(c_i x_i^*(h, N)^{-b_i} + A_i(Dh_i)^{-a_i} + E_i\big)$$
$$(4)$$

the data-dependent noise term $A_i(Dh_i)^{-a_i}$ breaks the symmetry with Eq. 1. Consequently, $h^*$ deviates from $w$, shifting weight toward domains that are "harder to learn" (larger $A_i$ and smaller $\alpha_i$).

We solve this bi-level problem using Online Mirror Descent (OMD) based on the following characterization:

**Proposition 4.2** (Gradient Characterization). *Let $x^*(h)$ and $\lambda(h)$ be the optimal solution and Lagrange multiplier of the capacity allocation problem (Eq. 1). The gradient of the outer objective $\mathcal{J}(h) := \sum_{i=1}^K w_i L_i(h, N, D)$ with respect to $h_k$ is:*

$$\nabla_k \mathcal{J}(h) = -w_k a_k A_k D^{-a_k} h_k^{-a_k-1} + \frac{\lambda x_k^*}{h_k(b_k+1)}\left(\bar{R} - \frac{w_k}{h_k}\right)$$

*where $\bar{R} = \dfrac{\sum_{j=1}^K \frac{x_j^*}{b_j+1}\left(\frac{w_j}{h_j}\right)}{\sum_{j=1}^K \frac{x_j^*}{b_j+1}}$.*

We defer the full algorithm to Section B.

**Tail Overlap.** As established in Section 4.2, overlapping skills contribute less to the fluctuation of $c_i x_i^*(h, N)^{-b_i}$ compared to disjoint skills. Similarly, an overlapping skill's aggregate observation count $D \cdot p(\text{skill})$ is more stable against mixture shifts, so its contribution to the noise term $A_i(Dh_i)^{-a_i}$ will be more stable as well. Thus, the overall loss fluctuation induced by overlapping skills remains small relative to disjoint skills.

## 5. Experiments

In this section, we empirically validate our proposed theoretical framework. We focus on three primary objectives:

1. **Predictive Accuracy:** We evaluate how well our theoretical model fits the observed loss landscape under various data mixtures compared to existing empirical baselines. Accuracy is assessed by the Mean Relative Error (MRE) and Mean Absolute Error (MAE).

2. **Optimal Mixture Identification:** We leverage the fitted scaling laws to predict the optimal training mixture $h^*$ and evaluate the test loss of the resulting models on the target distribution.

3. **Cross-Scale Extrapolation:** We test our model's ability to extrapolate to larger, unseen scales. Specifically, we fit our model parameters exclusively using small-scale test losses. We then substitute the target scale variables (e.g., model size $N$ and token budget $D$) into the fitted model to predict the optimal data mixture at the target scale. Finally, we empirically test this extrapolated mixture to validate its performance.

### 5.1. Experimental Setup

**Model and Data.** We first introduce the models and datasets used in our experiments.

- **Fitting Accuracy.** To evaluate predictive accuracy, we reuse the experimental data from (Liu et al., 2025a), which contain checkpoints for 64 1B-parameter models trained on different random mixtures. These 1B-parameter models were trained on 25B tokens spanning 17 domains from the train split of the Pile dataset (Gao et al., 2020) (see Table 1 in (Liu et al., 2025a) for full specifications). Their test losses are evaluated using the validation split of the same Pile dataset.[1] The models follow the TinyLlama architecture (Zhang et al.,

---

[1]For four domains where validation splits were unavailable in

*Table 2.* Comparison of fitting accuracy on 64 1B-parameter models trained on $K = 17$ domains from the Pile dataset. Our theoretically grounded models achieve the lowest error rates (MRE and MAE) while using significantly fewer total parameters than heuristic baselines.

| Method | MRE (%) ↓ | MAE ↓ | #Param |
|---|---|---|---|
| *Empirical Baselines* | | | |
| Additive | 2.209 | 0.052 | $K(2K+1)$ |
| Exponential | 6.990 | 0.059 | $K(K+2)$ |
| BiMix | 2.963 | 0.144 | $2K$ |
| RegMix | 6.480 | 0.136 | $K^2$ |
| *Our Models* | | | |
| Ours (Eq. (1)) | 2.064 | 0.051 | $3K$ |
| **Ours (Eq. 4)** | **1.533** | **0.034** | $5K$ |

2024), which is built on the Llama 2 tokenizer and structure, and were trained using a cosine schedule with a learning rate of $4 \times 10^{-4}$.

- **Optimal Mixture.** To evaluate the optimal data mixtures predicted by various scaling laws, we conduct experiments across three distinct settings: (1) A 4-domain configuration (Wikipedia, GitHub, StackExchange, and PG-19) adapted from Section 6 of (Shukor et al., 2026). Here, we train 25 domain mixtures—sampled from Pile-CC subsets—on 200M-parameter LLaMA-style models (Brown et al., 2020) for 8B tokens using a cosine learning rate schedule. (2) The primary experimental setup from Section 3 of (Shukor et al., 2026), which evaluates mixtures across varying model and token scales on the SlimPajama dataset with 7 domains using a constant learning rate. For this setting, we utilize the (122M parameters, 10B tokens) and (310M parameters, 20B tokens) configurations. (3) The aforementioned **17-domain** Pile setting (Liu et al., 2025a), where we optimize the training mixture to minimize test loss on the target Pile-CC distribution.

- **Cross-Scale Extrapolation.** To evaluate the extrapolation capabilities of our theoretical model, we conduct experiments across two distinct settings: (1) Using our 4-domain configuration (Wikipedia, GitHub, StackExchange, and PG-19), we extrapolate from a 200M-parameter model to a 700M-parameter model trained on 16B tokens. (2) Using the primary experimental setup from Section 3 of (Shukor et al., 2026) with 7-domains SlimPajama dataset, we extrapolate from the (122M parameters, 10B tokens) configuration to a larger (1B parameters, 30B tokens) configuration.

the original configuration (Liu et al., 2025a), we substitute the test loss with the training loss as a proxy.

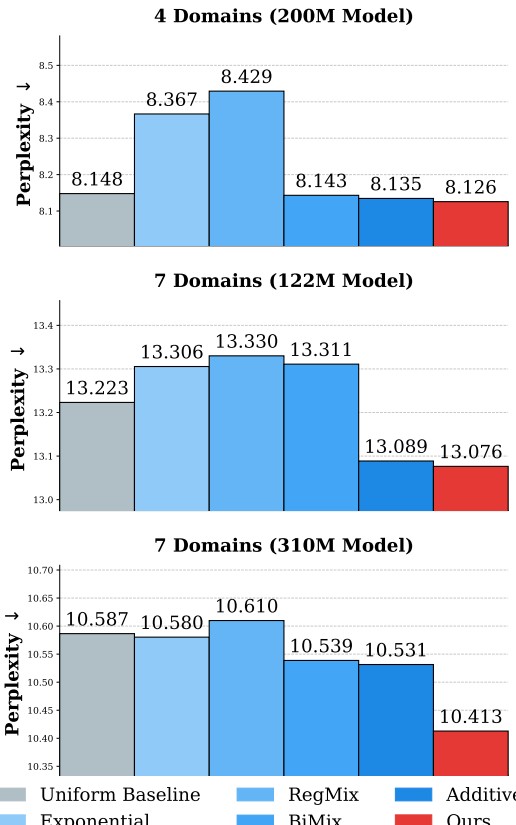

*Figure 2.* Performance comparison (test loss) of models trained with optimal mixtures predicted by different scaling laws. From left to right: 200M model on 4 domains, 122M model on 7 domains, and 310M model on 7 domains. Across all settings, our proposed method (highlighted in red) consistently achieves the lowest test loss compared to existing scaling laws, including exponential law (Ye et al., 2025), Regmix (Liu et al., 2025a), BiMix (Ge et al., 2025), and the additive law from (Shukor et al., 2026).

**Baselines and Metrics.** We benchmark our model against four empirical scaling laws from prior work, including (Shukor et al., 2026; Liu et al., 2025a; Ye et al., 2025; Ge et al., 2025). Fitting accuracy is evaluated via Mean Relative Error (MRE), while the quality of the predicted optimal mixtures is measured by the test loss and the perplexity.

### 5.2. Fitting Accuracy

**Procedure.** To evaluate each scaling law's ability to fit the loss landscape, we partition the 64 1B-parameter models, each trained on different mixtures, into a training set for fitting the scaling laws and a test set for assessing predictive accuracy. Goodness-of-fit is measured via the Mean Relative Error (MRE) between the predicted loss $\widehat{L}$ and the ground truth loss $L$ on the held-out mixtures, averaged over all domains. Formally, the MRE is defined as $\text{MRE} = \frac{1}{|\mathcal{H}_{test}| \cdot K} \sum_{h \in \mathcal{H}_{test}} \sum_{i=1}^{K} \left| \frac{\widehat{L}_i(h) - L_i(h)}{L_i(h)} \right|$, where

$\mathcal{H}_{test}$ is the set of held-out mixture weights.

**Fitting Methods.** To reliably fit the parameters of each scaling law, we select fitting methods based on the complexity of the model. For simple laws (power, exponential) such as (Ge et al., 2025; Ye et al., 2025), we estimate parameters using curve_fit from the scipy library, optimized with random starting points. For the Regmix method proposed by Liu et al. (2025a), we utilize LightGBM following the original implementation, setting the number of trees to $T = 100$ and the leaf size to $L = 31$. For intricate formulations like those in (Shukor et al., 2026), our Extended Quantization Model (with parameters $c_i, b_i, E_i$) and Extended Linear Model (with parameters $c_i, b_i, A_i, a_i, E_i$), we adopt the non-convex optimization strategy from (Shukor et al., 2026). This approach utilizes the Basin-Hopping algorithm (also from scipy) with L-BFGS as the inner routine to minimize the predictive error, optimized with random starting points as well. And when running L-BFGS for our models, we obtain the function value by solving the convex optimization (1) and utilize Mean Squared Error (MSE) as the objective function. Additionally, for laws containing terms like $(Nh_i)^{-a}$, we filter out $L_i$ with $h_i = 0$ to prevent division by zero.

**Results.** As shown in Table 2, our two proposed models achieve the lowest and second-lowest MRE (and MAE), with average MRE as low as $1.53\%$ and $2.06\%$. Notably, both models outperform the leading baseline (Shukor et al., 2026) while utilizing significantly fewer free parameters. This confirms that our theoretically grounded models— based on capacity allocation principles—capture the underlying mechanics of data mixing more accurately than purely heuristic curve-fitting.

### 5.3. Predicting the Optimal Data Mixture

We then test the ability of our Extended Linear Regression Model to predict the optimal training mixture $h^*$ that minimizes the loss on a target distribution, which is typically chosen as the uniform distribution over the $K$ domains, where $w = (\frac{1}{K}, \ldots, \frac{1}{K})$.

**Procedure.** We evaluate our Extended Linear Regression Model and the baselines across four distinct experiments (one from Setting (1), two from Setting(2), one from Setting (3)). For each experiment, we first fit the data mixing law using all baseline in Table 3, average mixture and our theoretical model. Once the laws are fitted, we compute the optimal mixture $h^*$ by solving the following optimization problem:

$$h^* = \arg\min_{h \in \Delta^{K-1}} \sum_{i=1}^{K} w_i \widehat{L}_i(h), \qquad (5)$$

where $\widehat{L}_i$ is the loss predicted by the fitted law. To solve this, we utilize standard solvers from cvxpy for convex baselines (e.g., the Exponential Law), and the L-BFGS algorithm with analytical gradients for non-convex formulations (including our Extended Linear Model and the Additive Law from (Shukor et al., 2026)). Finally, we train new models using the derived optimal mixtures $h^*$ and evaluate their performance across the four distinct experimental settings.

**Results.** Figure 2 presents the final test loss evaluated on the target distribution for the 200M (left), 122M (middle), and 310M (right) models. For our fourth setting on the 17-domain Pile dataset, Figure 5 illustrates the test loss on the Pile-CC domain. As observed, the mixture predicted by our Extended Linear Regression Model consistently yields the lowest test loss across all configurations. This demonstrates the superior accuracy and robustness of our approach in predicting optimal data mixtures compared to the uniform baseline and existing empirical laws. Crucially, we achieve this with significantly fewer free parameters.

### 5.4. Extrapolation to Unseen Scales

**Procedure.** To evaluate our model's extrapolation capabilities, we compare the performance of models trained at larger, unseen target scales using three distinct data mixtures. We conduct this evaluation across two experimental settings adapted from (Shukor et al., 2026): a 4-domain configuration extrapolating from a small scale of 200M/8B (parameters/tokens) to a target scale of 700M/16B, and a 7-domain configuration extrapolating from 122M/10B to 1B/30B. The three evaluated mixtures are:

1. **Our Static Mixture:** We first fit the parameters of our Extended Linear Regression Model ($c_i, b_i, A_i, a_i, E_i$) exclusively on the small-scale proxy regime. We then compute the optimal mixture by solving $h^*_{\text{static}} = \arg\min_{h \in \Delta^{K-1}} \sum_{i=1}^{K} w_i \widehat{L}_i(h, N, D)$ specifically for that same small scale.

2. **Our Extrapolated Mixture:** Using the exact same parameters fitted exclusively on the small-scale proxy regime, we derive the optimal data mixture for the unseen target scale. We achieve this by substituting $(N, D)$ with the target scales— $(700M, 16B)$ for the 4-domain setting and $(1B, 30B)$ for the 7-domain setting—and solving $h^*_{\text{extrapolated}} = \arg\min_{h \in \Delta^{K-1}} \sum_{i=1}^{K} w_i \widehat{L}_i(h, N, D)$.

3. **Baseline Mixtures (Target Scale):** We compare our mixtures against the state-of-the-art Additive Law from (Shukor et al., 2026). For the 4-domain setting, because the optimal mixture predicted by the Additive Law is theoretically scale-invariant, we use the mixture predicted by fitting the 200M/8B test losses.

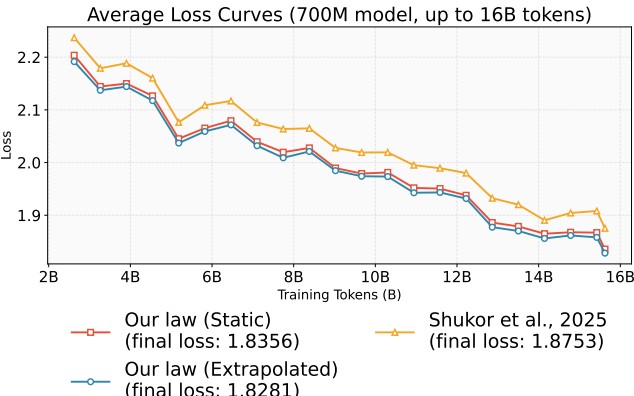

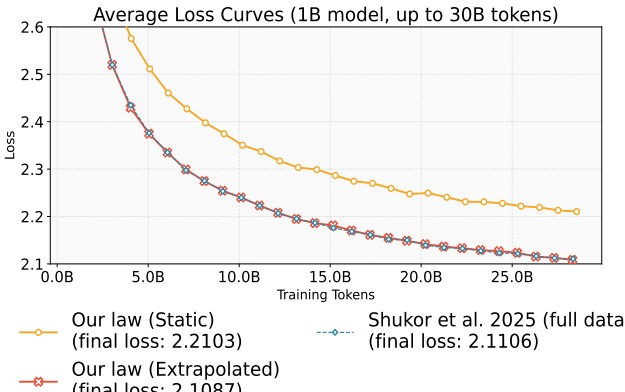

*Figure 3.* **Test loss throughout training for the 4-domain 200M/8B to 700M/16B extrapolation setting.** Our extrapolated mixture for the target 700M/16B scale strictly outperforms both the static mixture derived for the 200M/8B scale and the mixture predicted by the Additive Law (Shukor et al., 2026). This indicates that our framework correctly predicts the shift in the optimal data mixture as the scale increases.

For the 7-domain setting, we use the **full data optimal mixture** reported in Table 15 of (Shukor et al., 2026). This mixture acts as a strong baseline, as it was derived by fitting their Additive Law **across all available model and data sizes** ($N \in [412M, 1.4B]$, $D \in [4B, 46B]$)—crucially including the test losses from the target 1B/30B scale itself.

**Results.** We train models from scratch at the target scales using the aforementioned mixtures and evaluate their final performance.

For the 4-domain setting extrapolating to 700M/16B (Figure 3), our extrapolated mixture achieves the lowest test loss. Notably, the extrapolated mixture specifically adjusted for the 700M/16B target scale strictly outperforms the optimal mixture derived for the smaller 200M/8B scale, confirming that our framework successfully predicts how the optimal data mixture shifts as model size increases.

This advantage extends to the 7-domain 1B/30B extrapolation task (Figure 4). Most importantly, our framework—fitted exclusively with 122M/10B proxy losses—achieves the same final test loss as the state-of-the-art empirical baseline. This result is highly significant because the baseline is exceptionally strong, having been fitted on a wide spectrum of model and data sizes that explicitly includes the target 1B/30B scale. Matching this benchmark demonstrates that our law accurately predicts mixture shifts and reaches state-of-the-art performance at unseen large scales, relying entirely on small-scale proxy data.

*Figure 4.* **Test loss throughout training for the 7-domain 122M/10B to 1B/30B extrapolation setting.** Our extrapolated mixture strictly outperforms the optimal static mixture derived for the 122M/10B scale, correctly capturing the scale-dependent shift in the optimal data mixture. Most importantly, our extrapolated law—fitted *only* with 122M/10B losses—achieves the same test loss as the optimal mixture predicted by the state-of-the-art empirical law fitted with a massive range of model and data sizes (including the target 1B/30B scale itself). This demonstrates that our law not only accurately predicts mixture shifts, but also matches state-of-the-art performance at an unseen large scale using only small-scale proxy data.

# 6. Conclusion and Future Work

In this work, we proposed a theoretical model to explain data mixing scaling laws. Our framework accurately captures the loss landscape under various mixtures with low MRE and effectively identifies optimal mixtures. However, several promising directions remain for future exploration:

**Unseen Domains and Downstream Tasks.** Our current framework primarily addresses loss prediction within the span of training domains. A valuable future direction is to extend this framework to predict different data mixtures' effects on unseen domains or downstream tasks.

**Explicit Domain Overlap.** While the disjoint tail assumption appears effective in predicting the loss landscape, incorporating explicit modeling of information overlap may further refine predictions of the optimal mixture, particularly for highly correlated domains.

**Reliable Fitting Algorithms.** Fitting our model parameters currently requires solving a non-convex optimization problem, which can be computationally intensive and sensitive to initialization. A future direction is to develop more reliable estimation techniques, such as convex relaxations or analytical approximations.

# Acknowledgement

We gratefully acknowledge Kaifeng Lyu and Xinran Gu for valuable discussions, Pierre Ablin and Xiaosen Zheng for their clarification of experimental details and data, and the ICML reviewers for their insightful feedback.

# Impact Statement

This work is primarily theoretical in nature. We do not foresee any specific ethical concerns or immediate negative societal consequences arising from this study.

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

# A. Empirical Data Mixing Laws

As detailed in Table 3, these baselines offer distinct functional forms for predicting the domain loss $L_i(h)$ given the data mixture weights $h$. Specifically, we consider: (1) the **Additive Law** (Shukor et al., 2026), which models the loss using an inverse polynomial combination of mixture weights alongside model and data scale parameters ($N$ and $D$); (2) the **Exponential Law** (Ye et al., 2025), which assumes the loss decays exponentially based on a linear combination of the mixture weights; (3) **BiMix** (Ge et al., 2025), which factors the scaling effects of the target domain's weight against the total training tokens; and (4) **RegMix** (Liu et al., 2025a), which employs a simple yet effective linear regression directly over the mixture proportions.

*Table 3.* Comparison of empirical functional forms for predicting domain loss $L_i(h)$ based on mixture weights $h$. $N$ and $D$ represent model parameters and training tokens, respectively.

| Law Name | Functional Form ($f_i(h, N, D)$) |
|---|---|
| **Additive** (Shukor et al., 2026) | $L_i \approx E_i + \left( \sum_{j=1}^{K} C_{ij} h_j^{\gamma_{ij}} \right)^{-1} + \frac{A}{N^\alpha} + \frac{B}{D^\beta}$ |
| **Exponential** (Ye et al., 2025) | $L_i \approx c_i + k_i \exp\left( \sum_{j=1}^{K} t_{ij} h_j \right)$ |
| **BiMix** (Ge et al., 2025) | $L_i \approx \left( \frac{B}{D^\beta} + E \right) \frac{C}{h_i^\gamma}$ |
| **RegMix** (Liu et al., 2025a) | $L_i \approx w_0 + \sum_{j=1}^{K} w_j h_j$ |

# B. Algorithm

We propose Algorithm 1 with Exponentiated Gradient updates to find the optimal mixture $h^*$.

---

**Algorithm 1** Bi-Level Mixture Optimization via OMD

---

1: **Input:** Problem parameters, Target $w$, Constraints $N, D$, Step size $\eta$.
2: **Initialize:** $h^{(0)} \leftarrow [1/K, \ldots, 1/K]$, $t \leftarrow 0$.
3: **while** not converged **do**
4:     *// 1. Inner Level: Capacity Response*
5:     Solve the inner optimization (Eq. 1) given $h^{(t)}$ to obtain optimal allocation $x^*$ and multiplier $\lambda$.
6:     *// 2. Outer Level: Gradient Calculation*
7:     Compute the gradient vector $\nabla \mathcal{J}(h^{(t)})$ according to the closed-form solution in **Proposition 4.2**.
8:     *// 3. Optimization: Mirror Descent Step*
9:     Update weights (multiplicative): $\widetilde{h} \leftarrow h^{(t)} \odot \exp\left( -\eta \nabla \mathcal{J}(h^{(t)}) \right)$.
10:     Normalize: $h^{(t+1)} \leftarrow \widetilde{h}/\|\widetilde{h}\|_1$.
11:     $t \leftarrow t + 1$.
12: **end while**
13: **Return:** Optimal mixture $h^* \leftarrow h^{(t)}$.

---

# C. Implementation Details

For the experiments in Section 5.3, we implement our pretraining pipeline using the Megatron-LM framework in (Korthikanti et al., 2022) on a cluster of 8 NVIDIA H200 GPUs. To investigate the scaling laws with respect to data mixture ratios, we construct two GPT-style transformer models with varying scales: a 200M parameter model and a 700M parameter model.

**Model Architecture.** Both models follow the standard decoder-only transformer architecture. The 200M model consists of 24 layers, a hidden size of 768, and 12 attention heads. The 700M model scales the hidden size to 1536 while maintaining 24 layers and 12 heads. We utilize a sequence length of 1024 and employ Flash Attention to optimize training efficiency.

**Training Setup.** We train our models using the AdamW optimizer with $\beta_1 = 0.9$, $\beta_2 = 0.95$, and a weight decay of $0.1$. To isolate the effect of data mixture ratios, we adopt a constant learning rate schedule with a peak learning rate of $1.0 \times 10^{-4}$ and a 1% warmup phase, avoiding potential interference from cosine decay schedules during the mixture exploration. The global batch size is fixed at 512, achieved via a micro-batch size of 64 and gradient accumulation. Training is performed in BF16 precision. For 200M model, we train 8B tokens. For 700M model, we train 16B tokens.

**Data Configuration.** Our training corpus is composed of four domains: Github, Gutenberg, StackExchange, and Wikipedia. Since our primary focus is on data mixing laws, the mixture weights (ratios) of these domains vary across different experimental runs. Detailed hyperparameters for the model architecture and optimization are summarized in Table 4.

*Table 4.* Hyperparameters for the 200M and 700M proxy models used in our data mixing experiments.

| Hyperparameter | 200M Model | 700M Model |
|---|---|---|
| *Architecture* | | |
| Layers ($L$) | 24 | 24 |
| Hidden Size ($d_{model}$) | 768 | 1536 |
| Attention Heads | 12 | 12 |
| Sequence Length | 1024 | 1024 |
| *Optimization* | | |
| Global Batch Size | 128 | 128 |
| Micro Batch Size | 16 | 16 |
| Learning Rate | $3.5 \times 10^{-4}$ | $3.5 \times 10^{-4}$ |
| Minimum Learning Rate | $3.5 \times 10^{-5}$ | $3.5 \times 10^{-5}$ |
| LR Schedule | Cosine | Cosine |
| Warmup Ratio | 0.01 | 0.01 |
| Optimizer | AdamW | AdamW |
| Weight Decay | 0.1 | 0.1 |
| Precision | BF16 | BF16 |
| Training Tokens | 8B | 16B |

## D. Sketched Linear Regression

Following (Lin et al., 2024a; Bordelon et al., 2024; Li et al., 2026a), we consider a supervised learning setting where the goal is to learn a linear relationship from a stream of data. We analyze the dynamics of Stochastic Gradient Descent (SGD) under a *sketched observation* model, which characterizes the scenario where the model capacity (or feature access) is limited relative to the complexity of the data generating process. Our theoretical framework builds upon the work of Li et al. (2026a), who demonstrated a scaling law with respect to the learning rate schedule (LRS). In this work, we extend their analysis to derive the scaling law with respect to the mixture ratio $h$. Crucially, we no longer assume that the entire second-order moment of $\mathbf{x}$ follows a single power law; instead, the moment is partitioned into several blocks, with each block exhibiting distinct power-law scaling. In this section, we will compute the validation loss on each domain given a data mixture $h \in \Delta^{K-1}$. We investigate the linear model proposed in Section 4.3. Following the notation in Section 4.3, we use $\lambda_k^{(i)}$, to represent the k-th eigenvalue in domain $k$.

### D.1. Single-domain Framework

In the single-domain setting, (Bordelon et al., 2024; Lin et al., 2024a) provide a rigorous derivation of scaling laws by analyzing the training dynamics of linear regression under one-pass Stochastic Gradient Descent (SGD). In this framework, neural scaling laws are governed by the spectral decay of the data. As training progresses, the model "resolves" eigenmodes

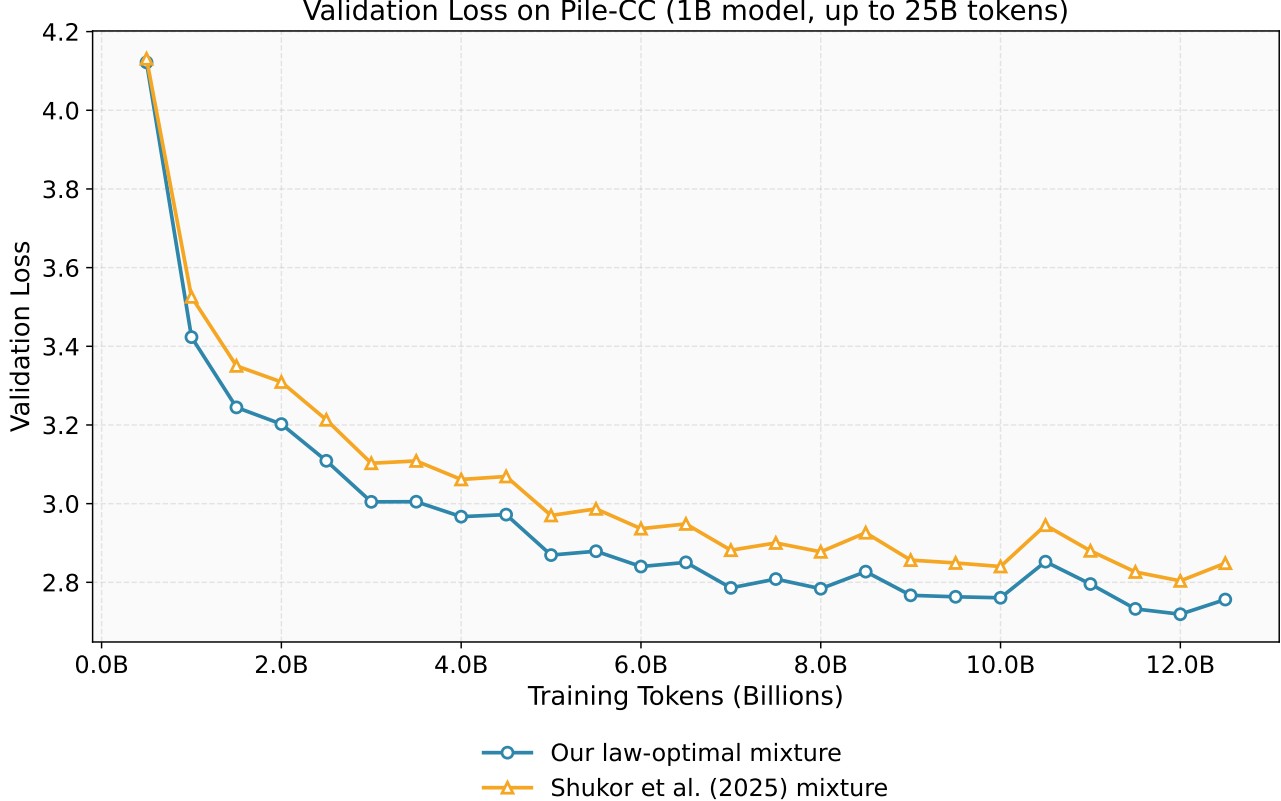

*Figure 5.* Test loss on Pile-CC. The model trained with the optimal mixture predicted by our fitted scaling law achieves a lower test loss compared to the mixture derived using the baseline law from Shukor et al. (2026).

in descending order of their eigenvalues—learning the dominant patterns first before fitting the fine-grained details. Below, we adopt the formal framework from (Lin et al., 2024a) to provide a simplified theoretical explanation.

**Data Generation:** Consider a linear regression problem where the input covariates $\mathbf{x} \in \mathbb{R}^d$ (where $d$ can be infinite) are drawn from a distribution with zero mean and covariance matrix $\mathbf{H} = \mathbb{E}[\mathbf{x}\mathbf{x}^\top]$. The target label $y$ is generated by a linear teacher with additive noise:

$$y = \langle \theta_*, \mathbf{x} \rangle + \epsilon,$$

where $\theta_*$ is the ground-truth parameter and $\epsilon \sim \mathcal{N}(0, \sigma^2)$ is independent Gaussian noise.

**Spectral Assumptions:** When training a linear regression model with one-pass SGD, the learning dynamics are determined by the spectrum of the covariance matrix $\mathbf{H} = \mathbb{E}[\mathbf{x}\mathbf{x}^\top]$. Intuitively, the eigenvalues $\lambda_k$ of $\mathbf{H}$ represent the variance (or signal strength) of the data along the $k$-th principal component. Empirical studies on natural data (e.g., images and text) consistently observe that these eigenvalues follow a power distribution (Field, 1987; Bahri et al., 2024). It is therefore assumed that the eigenvalues, sorted in descending order, follow a power law:

$$\lambda_k \propto k^{-\alpha}, \quad \text{for } \alpha > 1.$$

**Finite Parameter Projection:** To model a neural network with a finite capacity of $N$ parameters, we project the high-dimensional input $\mathbf{x}$ into a lower-dimensional feature space using a "sketching matrix" $\mathbf{S} \in \mathbb{R}^{N \times d}$. The model learns a weight vector $\theta \in \mathbb{R}^N$ by minimizing the squared error on the projected features $\widetilde{\mathbf{x}} = \mathbf{S}\mathbf{x}$:

$$\widehat{\theta} = \arg\min_{\theta \in \mathbb{R}^N} \frac{1}{D} \sum_{i=1}^{D} (y_i - \theta^\top \mathbf{S}\mathbf{x}_i)^2.$$

**Result:** Under this setup, (Lin et al., 2024a) derive the scaling law for the test loss $L(N, D)$ of the projected linear model $\widehat{w}$ (trained via one-pass SGD) as a function of the model size $N$ and training samples $D$:

$$L(N, D) \approx \underbrace{O\left(\frac{1}{N^{a_1}}\right)}_{\text{Model Scaling}} + \underbrace{O\left(\frac{1}{D^{a_2}}\right)}_{\text{Data Scaling}} + E.$$

### D.2. Multi-domain Framework

In this section, we formally describe the projected linear regression model in the multi-domain setting.

Following Section 3.1, we consider a linear regression problem over a union of $K$ domains. For each domain $i \in \{1, \dots, K\}$, input covariates $\mathbf{x}_i \in \mathbb{H}$ are feature vectors in a Hilbert space $\mathbb{H}$ (countably infinite-dimensional) drawn from a distribution with zero mean and a covariance matrix defined as $\mathbf{A}_i = \mathbb{E}_{\mathcal{P}_i}[\mathbf{x}_i \mathbf{x}_i^\top]$. The label $y$ is generated by a global linear teacher

$$y_i = \langle \theta^*, \mathbf{x}_i \rangle + \epsilon_i,$$

where $\theta^* \sim \mathcal{N}(0, \mathbf{I})$ is the ground-truth parameter and noise $\epsilon_i \sim \mathcal{N}(0, \sigma_i^2)$ is independent Gaussian noise.

Given a data mixture weight $h \in \Delta^{K-1}$, the input data $\mathbf{x} \in \mathbb{H}$ are drawn from a distribution $\mathcal{D} = \sum\limits_{i=1}^{K} h_i \mathcal{P}_i$ with zero mean and covariance matrix $\mathbf{H}(h) := \mathbb{E}_{\mathbf{x} \sim \mathcal{D}}[\mathbf{x}\mathbf{x}^\top] = \sum\limits_{i=1}^{K} h_i \mathbf{A}_i$, where $\mathbf{A}_i := \mathbb{E}_{\mathbf{x} \sim \mathcal{P}_i}[\mathbf{x}\mathbf{x}^\top]$. We assume that every domain contributes a non-zero proportion to data mixture.

**Assumption D.1.** There is a universal constant $\underline{h} > 0$ such that $h_i \geq \underline{h}$ holds for $i \in [K]$. For example, we can set $\underline{h} = 10^{-2}$.

#### D.2.1. DATA GENERATION

We make the following assumptions regarding the data distribution.

Firstly, we make the "Shared Head, Disjoint Tail" assumption. In spectral analysis, an eigenvector represents a specific direction or pattern of variation in the data (e.g., a specific texture in images or topic in text), while its corresponding eigenvalue quantifies the variance (or strength) of that pattern. Intuitively, we assume real-world data consists of *universal patterns* shared across all domains (the head) and *specialized nuances* unique to each domain (the tails). To make the analysis tractable, we assume that all covariance matrices $\{\mathbf{A}_i\}_{i=1}^{K}$ share a common orthonormal basis of eigenvectors $\mathbf{U} = [u_1, u_2, \dots]$ (i.e., they are simultaneously diagonalizable). In this basis, each matrix $\mathbf{A}_i$ is diagonal with eigenvalues denoted by $\lambda_k^{(i)}$. We classify these eigenvectors into two categories:

- **Shared Head ($k \leq H$):** The first $H$ eigenvectors $\{u_1, \dots, u_H\}$ represent universal components. We assume all domains possess non-zero variance along these directions ($\lambda_k^{(i)} > 0$ for all $i$), meaning these patterns are present in every domain.

- **Disjoint Tail ($k > H$):** The remaining eigenvectors represent domain-specific components. We assume each domain $i$ possesses a unique set of eigenvectors $\{u_k^{(i)}\}$ that are orthogonal to the specific components of other domains. Following Section 3.1, we assume that the variance along these directions follows a domain-specific power law:

$$u_k^{(i)\top} \mathbf{A}_j u_k^{(i)} = \begin{cases} k^{-\alpha_i} & \text{if } j = i \\ 0 & \text{if } j \neq i \end{cases}.$$

  Consequently, for $k > H$, the spectral structures are completely decoupled: each domain $i$ has positive eigenvalues $\lambda_k^{(i)} = k^{-\alpha_i}$ along its own unique eigenvectors and zero along the eigenvectors of others.

Under this assumption, the mixture covariance matrix $\mathbf{H}(h) = \sum h_j \mathbf{A}_j$ exhibits a decoupled structure in the tail. Because the domain-specific eigenvectors do not overlap, the eigenvalue of a unique component in the mixture is simply its

original variance scaled by the domain's proportion $h_j$. Specifically, for a tail eigenvector $u_k^{(i)}$ belonging to domain $i$, the corresponding eigenvalue in the mixture is:

$$\lambda(\mathbf{H}(h), u_k^{(i)}) = \sum_{j=1}^{K} h_j \cdot u_k^{(i)\top} \mathbf{A}_j u_k^{(i)} = h_i k^{-\alpha_i}.$$

In the following sections, we always work in basis $\mathbf{U} = \left[ u_1, u_2, \dots u_H, u_{H+1}^{(1)}, u_{H+1}^{(2)} \dots u_{H+1}^{(K)}, u_{H+2}^{(1)}, u_{H+2}^{(2)} \dots u_{H+2}^{(K)} \dots \right]$. For ease of notation, we absorb the domain-specific superscripts by flattening the doubly-indexed tail eigenvectors into a singly-indexed sequence, relabeling the set $\{u_k^{(i)} \mid k > H, i \in [K]\}$ sequentially as $u_{H+1}, u_{H+2}, \dots$. Formally, this establishes a mapping $\Gamma : \mathbb{N} \times \mathbb{N} \to \mathbb{N}$ for the tail indices ($k > H$ and $i \in [K]$), where the $k$-th index of domain $i$ is mapped to its new flattened index $\Gamma(i, k)$ via $\Gamma(i, k) = H + K(k - H - 1) + i$. In the following section, $\lambda_k$ denotes the eigenvalue associated with the eigenvector $u_k$, i.e. $\mathbf{H}u_k = \lambda_k u_k$ (but not the $k$-th largest eigenvalue). In this basis with flattened index, $A_i$ is diagonal with $\mathbf{A}_i = \mathrm{diag}(a_1, a_2, \dots)$ and

$$a_k = u_k^\top \mathbf{A}_i u_k = \begin{cases} l^{-\alpha_i} & k = \Gamma(i, l) \\ 0 & \text{otherwise.} \end{cases}$$

In addition, $\mathbf{H}(h) = \sum h_j \mathbf{A}_j$ is diagonal as well.

We then make a standard assumption regarding hypercontractivity, which is also used in (Li et al., 2026a). Intuitively, it requires the fourth moments of the data distribution to be reasonably bounded by its second moments, which ensures that the distribution $\mathcal{P}_i$ does not exhibit overly heavy tails.

**Assumption D.2** (Hypercontractivity). For any domain $i \in [K]$ and any positive semi-definite (PSD) matrix $\mathbf{M}$, there exists a constant $C_0 > 0$ such that

$$\mathbb{E}_{\mathbf{x} \sim \mathcal{P}_i} \left[ \mathbf{x}\mathbf{x}^\top \mathbf{M} \mathbf{x}\mathbf{x}^\top - \mathbf{A}_i \mathbf{M} \mathbf{A}_i \right] \preceq C_0 \mathrm{tr}(\mathbf{A}_i \mathbf{M}) \mathbf{A}_i.$$

We note that this is a very mild and standard assumption in theoretical analysis. A wide variety of standard distributions satisfy this condition for a small constant $C_0$, including but not limited to Gaussian distributions, sub-Gaussian distributions, and bounded distributions (such as uniform distributions on a sphere or a hypercube). Consequently, this assumption provides a robust foundation for concentration of measure without requiring strict distributional forms.

We assume the following regarding the prior of $\theta^*$.

**Assumption D.3.** Assume that $\theta^*$ satisfies a prior such that $\mathbb{E}[\theta^*]^{\otimes 2} = \mathbf{I}$.

This isotropic prior assumption is widely adopted in the theoretical analysis of overparameterized models (Lin et al., 2024a). Conceptually, it postulates that the optimal parameter $\theta^*$ exhibits no preferential directional bias in the feature space, meaning that the signal energy is uniformly distributed across all coordinate components. From a technical standpoint, this formulation isolates the effect of the parameter distribution, ensuring that the generalization error and the resulting scaling laws are fundamentally driven by the spectral properties of the data covariance matrix rather than the parameter alignment.

We assume the following regarding the range of power law exponents, based on empirical observations from (Kaplan et al., 2020; Hoffmann et al., 2022).

**Assumption D.4.** For all $i \in [K]$, $1 \leq \alpha_i \leq 3$.

### D.2.2. MODEL TRAINING

We then formalize the training process.

To model a neural network with a finite capacity of $N$ parameters, we project the high-dimensional input $\mathbf{x}$ into a lower-dimensional feature space using a **sketching operator** $\mathbf{S} \in \mathbb{R}^{N \times d}$. The model learns a weight vector $\theta \in \mathbb{R}^N$ by minimizing the squared error on the projected features $\widetilde{\mathbf{x}} = \mathbf{Sx}$:

$$\widehat{\theta} = \arg\min_{\theta \in \mathbb{R}^N} \frac{1}{D} \sum_{i=1}^{D} (y_i - \theta^\top \mathbf{S}\mathbf{x}_i)^2.$$

Now consider an arbitrary mixture $h \in \Delta^{K-1}$ and data covariance $\mathbf{H} = \sum_{i=1}^{K} h_i \mathbf{A}_i$. Following (Li et al., 2026a), we consider the *spectral truncation* sketch, which projects the input data onto the subspace spanned by the top-$N$ eigenvectors of $\mathbf{H}$. Formally,

$$\mathbf{S}_{i,j} = \begin{cases} 1, & \text{if } \lambda_j \text{ is the } i^{\text{th}} \text{ largest diagonal value,} \\ 0, & \text{otherwise,} \end{cases}$$

such that the eigenvalues of $\mathbb{E}[\widetilde{\mathbf{x}}\widetilde{\mathbf{x}}^\top] = \mathbb{E}[\mathbf{S}\mathbf{x}\mathbf{x}^\top\mathbf{S}^\top]$ are the largest $N$ eigenvalues of $\mathbf{H}$.

**Intuition behind S.** As discussed in Section 4.3, the eigenvalue $\mathbf{H}_{j,j}$ can be viewed as the frequency of the $j$-th skill. Therefore, by the top-$N$ sketching operator $\mathbf{S}$, the model essentially learns the $N$ skills with the highest frequencies.

Then the optimal model parameter $\widehat{\theta} = \arg\min_{\theta \in \mathbb{R}^N} \frac{1}{D}\sum_{i=1}^{D}(y_i - \theta^\top \mathbf{S}\mathbf{x}_i)^2$ is trained by one-pass SGD with cosine learning rate decay (Loshchilov & Hutter, 2017) on the sketched inputs. More specifically, we initialize the model parameter at $\theta_0 = \mathbf{0}$. At step $k$, the algorithm samples a data pair $(\mathbf{x}_k, y_k)$, constructs the observation $\widetilde{\mathbf{x}}_k = \mathbf{S}\mathbf{x}_k$, and performs a gradient descent update on the squared loss $\ell(\theta) = \frac{1}{2}(\widetilde{\mathbf{x}}_k^\top \theta - y_k)^2$:

$$\theta_{k+1} = \theta_k - \eta_k \nabla_\theta \ell(\theta_k; \widetilde{\mathbf{x}}_k, y_k)$$
$$= \theta_k - \eta_k \widetilde{\mathbf{x}}_k (\widetilde{\mathbf{x}}_k^\top \theta_k - y_k),$$

where $\eta_k = \eta_0 (1 + \cos(\pi k / D))$.

We assume that the training loss is always bounded.

**Assumption D.5.** The train loss over the training process has a finite upper bound $L$. Formally, for all $t$, $\mathbb{E}\left[(\langle \mathbf{S}\mathbf{x}, \theta(t) \rangle - y)^2\right] \leq L$.

We also impose an assumption on the parameter size $N$ and data size $D$:

**Assumption D.6.** There is a constant $\varepsilon > 0$ such that $N^{\alpha_i} \geq D^{1+\varepsilon}$ for all $i \in [K]$.

Assumption D.6 is a remarkably mild condition that is naturally satisfied in standard large-scale training regimes. Following empirical scaling laws (Kaplan et al., 2020), compute-optimal models typically scale the parameter size $N$ linearly with the dataset size $D$ (e.g., $D \approx 20N$). Under this linear scaling relationship ($N \propto D$), the left-hand side of our inequality scales as $\mathcal{O}(D^{\alpha_i})$. Since $\alpha_i > 1$ by definition, there strictly exists a sufficiently small constant $\varepsilon > 0$ such that $1 + \varepsilon \leq \alpha_i$. Consequently, the polynomial growth of $N^{\alpha_i}$ will trivially dominate $D^{1+\varepsilon}$ for large-scale $D$. Therefore, rather than imposing a restrictive capacity requirement, this assumption merely formalizes the standard overparameterized operational regime of modern language models, while simultaneously providing the necessary analytical bounds for our subsequent theoretical proofs.

### D.3. SDE Approximation

Following previous work (Li et al., 2026a; Qiu et al., 2025), we use continuous time approximation of one pass SGD, which simplify the analysis.

Since

$$\theta_{k+1} - \theta_k = -\eta_k \left(\mathbb{E}[\nabla_\theta \ell(\theta_k)] + \nabla_\theta \ell(\theta_k) - \mathbb{E}[\nabla_\theta \ell(\theta_k)]\right),$$

we have

$$\theta_D - \theta_0 = -\sum_{k=0}^{D-1} \eta_k \mathbb{E}[\nabla_\theta \ell(\theta_k)] - \sum_{k=0}^{D-1} \eta_k \left(\nabla_\theta \ell(\theta_k) - \mathbb{E}[\nabla_\theta \ell(\theta_k)]\right).$$

We generalize the discrete sequence $\{\theta_0, \ldots, \theta_D\}$ to a continuous function $\theta(\cdot)$, and similarly extend $\eta_k$ to $\eta(\cdot)$.

Now we compute $\mathbf{\Sigma}(\theta_k) := \left(\nabla_\theta \ell(\theta_k) - \mathbb{E}[\nabla_\theta \ell(\theta_k)]\right)^{\otimes 2}$.

Since

$$\nabla_\theta \ell(\theta_k) = \mathbf{S}\mathbf{x}_k \mathbf{x}_k^\top \left(\mathbf{S}^\top \theta_k - \theta^*\right) - \epsilon_k \mathbf{S}\mathbf{x}_k,$$

we have

$$\boldsymbol{\Sigma}(\theta_k) = \mathbb{E}\left[(\nabla_\theta \ell(\theta_k) - \mathbb{E}[\nabla_\theta \ell(\theta_k)])\right]^{\otimes 2} = \left[\mathbf{S}\left(\mathbf{x}_k \mathbf{x}_k^\top - \mathbf{H}\right)\left(\mathbf{S}^\top \theta - \theta^*\right)\right]^{\otimes 2} + \left(\sum_{i \in [K]} h_i \sigma_i^2\right)\mathbf{SHS}^\top.$$

Using Euler-Maclaurin equation, we have

$$\theta_D - \theta_0 \approx \theta(D) - \theta(0) = \int_0^D -\eta(k)\mathbf{SH}\left(\mathbf{S}^\top \theta(k) - \theta^*\right)\mathrm{d}k + \int_0^D \eta(k)\sqrt{\boldsymbol{\Sigma}(\theta(k))}\mathrm{d}\mathbf{B}_k, \tag{6}$$

where $\mathbf{B}_k \in \mathbb{R}^N$ is a N-dimensional Brownian motion. Following (Li et al., 2026a), we define **intrinsic time**

$$\tau(t) := \int_0^t \eta(k)\mathrm{d}k.$$

We can rewrite equation 6 as

$$\theta(D) - \theta(0) = \int_0^D -\mathbf{SH}(\mathbf{S}^\top \theta(k) - \theta^*)\mathrm{d}\tau(k) + \sqrt{\eta(k)\boldsymbol{\Sigma}(\theta(k))}\mathrm{d}\mathbf{B}_\tau. \tag{7}$$

Taking the derivative of Equation 7, we have the following lemma.

**Lemma D.7.** *Define* $\mathbf{w}(t) := \mathbf{S}^\top \theta(\tau^{-1}(t)) - \theta^*, \gamma(t) := \eta(\tau^{-1}(t))$, *we have*

$$\mathrm{d}\mathbf{w} = -\mathbf{S}^\top \mathbf{SHw}\mathrm{d}t + \mathbf{S}^\top \sqrt{\gamma(t)\boldsymbol{\Sigma}(\theta(\tau^{-1}(t)))}\mathbf{S}\mathrm{d}\mathbf{B}_t,$$

*where* $\mathbf{B}_t \in \mathbb{R}^d$ *is a d-dimensional Brownian motion.*

## D.4. Test Loss

We first compute the expected test loss on a certain domain $\tau \in [K]$ when the model is trained with mixture $h \in \Delta^{K-1}$.

In our projected linear regression framework, the test loss on domain $\tau$ equals to

$$\begin{aligned} L_\tau &= \mathbb{E}_{\mathbf{x} \sim \mathcal{P}_\tau}\left[(\langle \mathbf{Sx}, \theta\rangle - \langle \mathbf{x}, \theta^*\rangle)^2\right] + \sigma^2 \\ &= \mathbb{E}\left[\mathbf{w}^\top(D\eta_0)\mathbf{A}_\tau \mathbf{w}(D\eta_0)\right] + \sigma^2. \end{aligned}$$

**Theorem D.8.** *Consider the projected linear regression model defined in Section D.2 and consider an arbitrary domain* $\tau$. *Let* $r_k$ *be the rank of the eigenvalue* $\lambda_k$ *among all eigenvalues of* $\mathbf{H} = \sum h_j \mathbf{A}_j$, *sorted in descending order. Let* $\mathbf{A}_\tau = diag(a_1, a_2, \dots)$ *be domain* $\tau$*'s data covariance (i.e.* $\mathbf{A}_\tau = \mathbb{E}_{\mathcal{P}_\tau}[\mathbf{xx}^\top]$*) and*

$$a_k = \begin{cases} l^{-\alpha_\tau} & k = \Gamma(\tau, l) \\ 0 & otherwise \end{cases}$$

*then the expected test loss on domain* $\tau$, *denoted by* $L_\tau$, *satisfies*

$$Bias + Approx + Var_1 + \sigma_\tau^2 \le L_\tau \le Bias + Approx + Var_2 + \sigma_\tau^2, \tag{8}$$

*where*

$$Bias := \sum_{k: r_k \le N} \exp(-2\lambda_k D\eta_0)a_k,$$

$$Approx := \sum_{k: r_k > N} a_k,$$

$$Var_1 := \sum_{k: r_k \le N}\left(\sum_{i \in [K]} h_i \sigma_i^2\right) a_k \lambda_k \int_0^{D\eta_0} \exp(-2\lambda_k(D\eta_0 - t))\gamma(t)\mathrm{d}t,$$

$$Var_2 := \sum_{k: r_k \le N} a_k \lambda_k \int_0^{D\eta_0} \exp(-2\lambda_k(D\eta_0 - t))\gamma(t)\left(\sum_{i \in [K]} h_i \sigma_i^2 + \frac{C_0}{\underline{h}}\mathbb{E}\left[\mathbf{w}(t)^\top \mathbf{Hw}(t)\right]\right)\mathrm{d}t,$$

*and* $\gamma(t) = \eta(\tau^{-1}(t)), \lambda_k = u_k^\top \mathbf{H}u_k = \mathbf{H}_{k,k}$.

*Proof.* We consider the quadratic function $f(\mathbf{w}) = \mathbf{w}^\top \mathbf{A}\mathbf{w}$ for any diagonal matrix $\mathbf{A} = \text{diag}(a_1', a_2' \ldots)$ (not necessarily $\mathbf{A}_i$).Recall that for a multidimensional stochastic process, Itô's Lemma expands a twice-differentiable function $f(\mathbf{w})$ as:

$$\mathrm{d}f(\mathbf{w}(t)) = (\nabla f(\mathbf{w}(t)))^\top \mathrm{d}\mathbf{w}(t) + \frac{1}{2}\text{tr}\left[\nabla^2 f(\mathbf{w}(t)) \cdot (\mathrm{d}\mathbf{w}(t))^{\otimes 2}\right]$$

For our specific function, the gradient is $\nabla f = 2\mathbf{A}\mathbf{w}$ and the Hessian is $\nabla^2 f = 2\mathbf{A}$. Substituting these derivatives into Itô's formula and taking the expectation yields:

$$\mathrm{d}\mathbb{E}[\mathbf{w}^\top(t)\mathbf{A}\mathbf{w}(t)] = \mathbb{E}\left[2\mathbf{w}^\top(t)\mathbf{A}\mathrm{d}\mathbf{w}(t) + \frac{1}{2}\text{tr}\left[2\mathbf{A}\cdot(\mathrm{d}\mathbf{w}(t))^{\otimes 2}\right]\right]$$

$$= \mathbb{E}\left[-2\mathbf{w}^\top(t)\mathbf{A}\mathbf{S}^\top\mathbf{S}\mathbf{H}\mathbf{w}\mathrm{d}t + \gamma(t)\mathbf{w}^\top(\mathbf{x}\mathbf{x}^\top - \mathbf{H})\mathbf{S}^\top\mathbf{A}\mathbf{S}(\mathbf{x}\mathbf{x}^\top - \mathbf{H})\mathbf{w}\mathrm{d}t + \gamma(t)\left(\sum_{i\in[K]}h_i\sigma_i^2\right)\text{tr}[\mathbf{A}\mathbf{S}\mathbf{H}\mathbf{S}^\top]\mathrm{d}t\right].$$

We first take expectation over $\mathbf{x}$ with the help of Assumption D.2 and Assumption D.1, for the term $\mathbf{w}^\top(\mathbf{x}\mathbf{x}^\top - \mathbf{H})\mathbf{S}^\top\mathbf{A}\mathbf{S}(\mathbf{x}\mathbf{x}^\top - \mathbf{H})\mathbf{w}$, we have

$$\mathbb{E}_{\mathbf{x}\sim\mathcal{D}(h)}\left[\mathbf{w}^\top(\mathbf{x}\mathbf{x}^\top - \mathbf{H})\mathbf{S}^\top\mathbf{A}\mathbf{S}(\mathbf{x}\mathbf{x}^\top - \mathbf{H})\mathbf{w}\right] = \mathbf{w}^\top\mathbb{E}\left[\mathbf{x}\mathbf{x}^\top\mathbf{S}^\top\mathbf{A}\mathbf{S}\mathbf{x}\mathbf{x}^\top - \mathbf{H}\mathbf{S}^\top\mathbf{A}\mathbf{S}\mathbf{H}\right]\mathbf{w}$$

$$= \mathbf{w}^\top\sum h_i\mathbb{E}_{\mathbf{x}\sim\mathcal{P}_i}\left[\mathbf{x}\mathbf{x}^\top\mathbf{S}^\top\mathbf{A}\mathbf{S}\mathbf{x}\mathbf{x}^\top - \mathbf{A}_i\mathbf{S}^\top\mathbf{A}\mathbf{S}\mathbf{A}_i\right]\mathbf{w}$$

$$\leq \mathbf{w}^\top\sum h_iC_0\text{tr}[\mathbf{S}^\top\mathbf{A}\mathbf{S}\mathbf{A}_i]\mathbf{A}_i\mathbf{w}$$

$$\leq \mathbf{w}^\top\frac{C_0}{\underline{h}}\sum\text{tr}[\mathbf{S}^\top\mathbf{A}\mathbf{S}\mathbf{A}_i]h_i\mathbf{A}_i\mathbf{w}$$

$$= \frac{C_0}{\underline{h}}\text{tr}[\mathbf{S}^\top\mathbf{A}\mathbf{S}\mathbf{H}]\mathbf{w}(t)^\top\mathbf{H}\mathbf{w}(t).$$

Set $\mathbf{A}$ as $\mathbf{E}_{k,k}$, let $y_k(t) = \mathbb{E}[\mathbf{w}_k^2(t)]$, we can get the two side bound with respect to $y_k$:

- For $r_k \leq N$,

$$-2\lambda_k y_k(t) + \gamma(t)\lambda_k \cdot \left(\sum_{i\in[K]}h_i\sigma_i^2\right) \leq y_k'(t) \leq -2\lambda_k y_k(t) + \gamma(t)\lambda_k\left(\sum_{i\in[K]}h_i\sigma_i^2 + \frac{C_0}{\underline{h}}\mathbb{E}[\mathbf{w}(t)^\top\mathbf{H}\mathbf{w}(t)]\right).$$

- For $r_k > N$, $y_k'(t) = 0$.

Solving the differential equation, for all $k$ such that $r_k \leq N$, we have

$$y_k(D\eta_0) \geq \exp(-2\lambda_k D\eta_0)y_k(0) + \left(\sum_{i\in[K]}h_i\sigma_i^2\right)\cdot\lambda_k\int_0^{D\eta_0}\exp(-2\lambda_k(D\eta_0 - t))\gamma(t)\mathrm{d}t,$$

$$y_k(D\eta_0) \leq \exp(-2\lambda_k D\eta_0)y_k(0) + \lambda_k\int_0^{D\eta_0}\exp(-2\lambda_k(D\eta_0 - t))\gamma(t)\left(\left(\sum_{i\in[K]}h_i\sigma_i^2\right) + \frac{C_0}{\underline{h}}\mathbb{E}[\mathbf{w}(t)^\top\mathbf{H}\mathbf{w}(t)]\right)\mathrm{d}t.$$

By $\mathbb{E}[\mathbf{w}^\top(t)\mathbf{A}\mathbf{w}(t)] = \sum y_k(t)a_k$ and $\mathbb{E}[\mathbf{w}_i^2(0)] = 1$(by Assumption D.3), we have

$$\mathbb{E}[\mathbf{w}^\top(D\eta_0)\mathbf{A}\mathbf{w}(D\eta_0)] = \sum y_k(D\eta_0)a_k$$

$$\geq \sum_{k:r_k>N}a_k$$

$$+ \sum_{k:\,r_k\leq N}\exp(-2\lambda_k D\eta_0)a_k$$

$$+ \sum_{k:r_k\leq N}\left(\sum_{i\in[K]}h_i\sigma_i^2\right)a_k\lambda_k\int_0^{D\eta_0}\exp(-2\lambda_k(D\eta_0 - t))\gamma(t)\mathrm{d}t$$

and

$$
\begin{aligned}
\mathbb{E}[\mathbf{w}^\top(D\eta_0)\mathbf{A}\mathbf{w}(D\eta_0)] &= \sum y_k(D\eta_0)a_k \\
&\leq \sum_{k:r_k>N} a_k \\
&+ \sum_{k:\,r_k\leq N} \exp(-2\lambda_k D\eta_0)a_k \\
&+ \sum_{k:\,r_k\leq N} a_k\lambda_k \int_0^{D\eta_0} \exp(-2\lambda_k(D\eta_0-t))\gamma(t)\left(\sum_{i\in[K]} h_i\sigma_i^2 + \frac{C_0}{\underline{h}}\mathbb{E}\left[\mathbf{w}(t)^\top\mathbf{H}\mathbf{w}(t)\right]\right)\mathrm{d}t.
\end{aligned}
$$

$\square$

In the following sections, we estimate each term in the bounds separately.

## D.5. The Approximation Error Term

Consider an arbitrary domain $\tau$, we first provide a bound for the approximation error term $\text{Approx} = \sum_{k:\,r_k>N} a_k$. We demonstrate that, aside from a gap introduced by discretization, this term is equivalent to the domain loss $L_i(h) = c_i x_i^*(h)^{-b_i}$ in our Extended Quantization Model, with $c_i = \frac{1}{\alpha_i-1}$ and $b_i = \alpha_i - 1$.

**Theorem D.9.** *Consider the projected linear regression model defined in Section D.2. Let $x^*$ be the solution to the Problem 1 with parameter $c_i = \frac{1}{\alpha_i-1}, b_i = \alpha_i - 1$. Then for any domain $\tau$, the term Approx in Theorem D.8 satisfies*

$$
\left|\text{Approx} - c_\tau(x_\tau^*)^{-b_\tau}\right| \leq \frac{C_{13}}{N^{\alpha_{\min}}}
$$

*for sufficiently large $N$ and $C_{13}$ is a constant that only depends on $\alpha$.*

### D.5.1. PROOF OF THEOREM D.9

To understand why the term Approx is equivalent to the domain loss in the Extended Quantization Model for an arbitrary domain $\tau$, note that Theorem D.8 establishes:

$$
\text{Approx} = \sum_{k:\,r_k>N} a_k,
$$

where $r_k$ is the rank of the eigenvalue $\lambda_k$ among all eigenvalues of $\mathbf{H} = \sum h_j\mathbf{A}_j$, sorted in descending order, and

$$
a_k = \begin{cases} l^{-\alpha_\tau} & k = \Gamma(\tau,l) \\ 0 & \text{otherwise.} \end{cases}
$$

If we interpret the eigenvalues as the frequencies of skills, this formulation indicates that Approx is precisely the sum of the frequencies of the unlearned skills in domain $\tau$, assuming the model learns the $N$ skills with the highest frequencies (under a training mixture $h$). This aligns perfectly with the Extended Quantization Model if we define $c_\tau$ as the normalization constant such that $\sum_k a_k/c_\tau = 1$. Under this formulation, the optimization objective (Problem 1) minimizes the total loss by learning the highest-frequency skills across all $K$ domains, and the loss for any domain $\tau$ is incurred by its unlearned skills. However, since the Extended Quantization Model considers a continuous skill space, there will be a gap due to discretization.

We first prove a lower bound for $x_\tau^*$ as a function of $N$, ensuring that $x_\tau^*$ grows strictly monotonically with $N$. The lemma also guarantees the asymptotic relationship $\text{Approx} = \frac{(x_\tau^*)^{1-\alpha_\tau}}{\alpha_\tau-1}(1 + o_N(1))$.

**Lemma D.10.** *Consider the Extended Quantization Model in Section 4.2 (Problem 1):*

$$\min_{\mathbf{x}} \quad L(\mathbf{x}) = \sum_{i=1}^{K} h_i c_i x_i^{-b_i}$$

$$\text{s.t.} \quad \sum_{i=1}^{K} (x_i - H) \leq N - H, \tag{9}$$

$$x_i \geq H, \quad \forall i \in \{1, \ldots, K\},$$

*where $h_i, c_i, b_i > 0$ for all $i$. Let $b_{\min} = \min_{1 \leq i \leq K} b_i$. As $N \to \infty$, the optimal solution $x^*$ satisfies the asymptotic lower bound:*

$$x_\tau^* = \Omega\left(N^{\frac{b_{\min}+1}{b_\tau+1}}\right).$$

*Proof.* Let $A_i = h_i c_i > 0$ for all $i \in \{1, \ldots, K\}$. Since $A_i > 0$ and $b_i > 0$, the objective function $L(\mathbf{x})$ is strictly monotonically decreasing with respect to each variable $x_i$. Consequently, to minimize the objective function, the variables $x_i$ must take the largest possible values permitted by the feasible region. This implies that the sum constraint must be active at the optimal solution, yielding the equality $\sum_{i=1}^{K} x_i = N + (K-1)H$. Furthermore, as the total available resource $N$ approaches infinity, the optimal values $x_i^*$ will also approach infinity. Thus, for sufficiently large $N$, the lower bound constraints $x_i \geq H$ become strictly inactive and can be omitted from the asymptotic analysis.

We proceed by applying the method of Lagrange multipliers. The Lagrangian associated with the equality constraint is given by

$$\mathcal{L}(\mathbf{x}, \lambda) = \sum_{i=1}^{K} A_i x_i^{-b_i} + \lambda\left(\sum_{i=1}^{K} x_i - N - (K-1)H\right),$$

where $\lambda > 0$ is the Lagrange multiplier. Taking the partial derivative of $\mathcal{L}$ with respect to $x_i$ and equating it to zero yields the first-order necessary conditions for optimality:

$$\frac{\partial \mathcal{L}}{\partial x_i} = -b_i A_i x_i^{-(b_i+1)} + \lambda = 0, \quad \forall i \in \{1, \ldots, K\}.$$

Rearranging this expression, we obtain a relationship between the optimal variable $x_i$ and the multiplier $\lambda$:

$$\lambda = b_i A_i x_i^{-(b_i+1)}.$$

Since $\lambda$ is a global constant across all dimensions, we can equate the expressions for an arbitrary index $i$ and the specific index $\tau$, yielding

$$b_i A_i x_i^{-(b_i+1)} = b_\tau A_\tau x_\tau^{-(b_\tau+1)}.$$

Solving this equation for $x_i$ in terms of $x_\tau$, we find

$$x_i = \left(\frac{b_i A_i}{b_\tau A_\tau}\right)^{\frac{1}{b_i+1}} x_\tau^{\frac{b_\tau+1}{b_i+1}}.$$

Substituting this relationship back into the active resource constraint gives

$$\sum_{i=1}^{K} \left(\frac{b_i A_i}{b_\tau A_\tau}\right)^{\frac{1}{b_i+1}} x_\tau^{\frac{b_\tau+1}{b_i+1}} = N + (K-1)H.$$

We now analyze the asymptotic behavior of this equation as $N \to \infty$. On the right-hand side, the constant term $(K-1)H$ becomes negligible, so the right-hand side is asymptotically equivalent to $N$. On the left-hand side, we have a sum of fractional powers of $x_\tau$. As $N \to \infty$ implies $x_\tau \to \infty$, the behavior of the sum is completely dominated by the term with the highest exponent. The exponent of $x_\tau$ for the $i$-th term is $\frac{b_\tau+1}{b_i+1}$. This exponent is maximized when its denominator, $b_i + 1$, is minimized, which occurs exactly when $b_i = b_{\min} = \min_{1 \leq j \leq K} b_j$.

Let $\mathcal{I}_{\min} = \{i \mid b_i = b_{\min}\}$ be the index set of all terms achieving this minimum exponent. Extracting these dominant terms, we establish the asymptotic equivalence

$$\sum_{i \in \mathcal{I}_{\min}} \left(\frac{b_{\min} A_i}{b_\tau A_\tau}\right)^{\frac{1}{b_{\min}+1}} x_\tau^{\frac{b_\tau+1}{b_{\min}+1}} \sim N.$$

Letting $C = \sum_{i \in \mathcal{I}_{\min}} \left(\frac{b_{\min} A_i}{b_\tau A_\tau}\right)^{\frac{1}{b_{\min}+1}}$, which is a strictly positive constant, the relation simplifies to

$$C x_\tau^{\frac{b_\tau+1}{b_{\min}+1}} \sim N.$$

Solving this asymptotic equivalence for $x_\tau$ yields

$$x_\tau \sim \left(\frac{1}{C}\right)^{\frac{b_{\min}+1}{b_\tau+1}} N^{\frac{b_{\min}+1}{b_\tau+1}}.$$

This demonstrates that the growth rate of $x_\tau^*$ is proportional to $N^{\frac{b_{\min}+1}{b_\tau+1}}$. Therefore, we conclude that the optimal solution $x_\tau^*$ satisfies the strict asymptotic lower bound $x_\tau^* = \Omega\left(N^{\frac{b_{\min}+1}{b_\tau+1}}\right)$, completing the proof. $\square$

We are now ready to prove Theorem D.9. We prove the equivalence of Approx and the domain loss under the Extended Quantization Model, and we bound the discretization gap as follows.

By setting $c_i = \frac{1}{\alpha_i - 1}$, Problem 1 can be written as:

$$\min_x \quad L = \sum_{i=1}^{K} h_i \frac{1}{\alpha_i - 1} x_i^{1-\alpha_i}$$

$$\text{s.t.} \quad \sum_{i=1}^{K}(x_i - H) \leq N - H,$$

$$x_i \geq H, \quad \forall i.$$

Let $x^*$ be the optimal continuous solution of the optimization problem above, and let $x'_\tau := \max\{k : r_{\Gamma(\tau,k)} \leq N\}$ be the number of eigenvalues from domain $\tau$ that are selected by the sketching operator $\mathbf{S}$ (with the shared head counted as well). We are to bound the difference in $x_\tau^*$ and $x'_\tau$ as $|x_\tau^* - x'_\tau| \leq K$, which is tight enough to obtain the conclusion in this theorem.

**Lemma D.11.** *For any domain $\tau$, $|x_\tau^* - x'_\tau| \leq K$, where $K$ is the number of domains.*

*Proof.* We first characterize the optimal continuous solution $x^*$ via KKT conditions. By the KKT conditions, there exist multipliers $\lambda \geq 0$ and $\mu_i \geq 0$ such that:

$$\sum_{i=1}^{K} x_i^* = N + (K-1)H, \tag{10}$$

$$\frac{h_i}{(x_i^*)^{\alpha_i}} - \mu_i = \lambda, \quad \forall i, \tag{11}$$

$$\mu_i(x_i^* - H) = 0, \quad \forall i. \tag{12}$$

Note that equality holds in (10) because the objective function strictly decreases as $x_i$ increases.

By Lemma D.10, for sufficiently large $N$, we have $x_i^* > H$ for all $i \in [K]$, which by Equation (12) implies that $\mu_i = 0$ for all $i \in [K]$. As a result, by Equation (11), we have

$$\frac{h_i}{(x_i^*)^{\alpha_i}} = \lambda, \quad \forall i$$

that is,

$$x_i^* = \left(\frac{h_i}{\lambda}\right)^{\frac{1}{\alpha_i}}, \quad \forall i.$$

Then by the capacity constraint Equation (10), we easily find the optimal $x^*$ by finding the $\lambda$ that satisfies the capacity constraint. Define $S(\lambda) := \sum_{i \in [K]} \left(\frac{h_i}{\lambda}\right)^{\frac{1}{\alpha_i}}$. It is easy to verify that the equation $S(\lambda) = N + (K-1)H$ has a unique solution $\lambda^*$. Then for all $i \in [K]$,

$$h_i(x_i^*)^{-\alpha_i} = \lambda^*.$$

Next we bound the difference between $x_\tau^*$ and $x_\tau'$, where $x_\tau'$ represents the number of eigenvalues from domain $\tau$ that are selected by the sketching operator $\mathbf{S}$ (with the shared head counted). Recall that the sketching operator $\mathbf{S}$ picks the largest $N$ eigenvalues and the $k$-th largest eigenvalue from domain $i$ equals $h_i k^{-\alpha_i}$. We connect $x_\tau^*$ and $x_\tau'$ by picking eigenvalues instead by a common threshold $\lambda^* = h_i(x_i^*)^{-\alpha_i}$. Define

$$U = \{j : \mathbf{H}_{j,j} \geq \lambda^*\}.$$

Then $U$ must pick the first $y_i = \lfloor x_i^* \rfloor$ eigenvalues from domain $i$ for all $i$ (with the shared head counted). Since $x_i^* - 1 < y_i \leq x_i^*$ and $H + \sum_{i=1}^{K}(x_i^* - H) = N$, we must have the following for the size of $U$:

$$N - K < |U| = \left(H + \sum_{i=1}^{K}(y_i - H)\right) \leq N.$$

Thus, the threshold $\lambda^* = h_i(x_i^*)^{-\alpha_i}$ selects at least $N - K + 1$ eigenvalues, and these eigenvalues must be the largest ones. This implies that the sketching operator $\mathbf{S}$ must select all eigenvalues in $U$, while adding at most $K - 1$ additional eigenvalues from any given domain $\tau$. Therefore, we have:

$$x_\tau^* - K \leq y_\tau \leq x_\tau' \leq y_\tau + (K-1) \leq x_\tau^* + K.$$

$\square$

We can now bound the difference between Approx and $\frac{(x_\tau^*)^{1-\alpha_\tau}}{\alpha_\tau - 1}$. By Approx $= \sum_{k:r_k \leq N} a_k = \sum_{j > x_\tau'} j^{-\alpha_\tau}$, we have

$$\sum_{j > y_\tau + K} j^{-\alpha_\tau} \leq \text{Approx} \leq \sum_{j > y_\tau} j^{-\alpha_\tau}$$

To bound these discrete sums, define the continuous integral tail $F_\tau(z) = \int_z^\infty x^{-\alpha_\tau} \mathrm{d}x = \frac{z^{1-\alpha_\tau}}{\alpha_\tau - 1}$. Bounding the sums with integrals gives:

$$F_\tau(y_\tau + K + 2) \leq \text{Approx} \leq F_\tau(y_\tau)$$

Because $y_\tau \leq x_\tau^* < y_\tau + 1$, the deviation between the discrete approximation and the continuous ideal $F_\tau(x_\tau^*)$ is constrained by the maximum index gap. Therefore, we have:

$$\left| \text{Approx} - \frac{(x_\tau^*)^{1-\alpha_\tau}}{\alpha_\tau - 1} \right| \leq \frac{K+2}{(x_\tau^*)^{\alpha_\tau}}$$
$$\leq \frac{C_{13}}{N^{\alpha_{\min}}}$$

where the final inequality follows from Lemma D.10, and $C_{13}$ is a constant depending only on $\alpha$. This completes the proof of Theorem D.9.

## D.6. The Bias Term

In this section, we bound the bias term $\text{Bias} = \sum\limits_{k:\, r_k \leq N} \exp(-2\lambda_k D\eta_0) a_k$ where

$$a_k = \begin{cases} l^{-\alpha_\tau} & k = \Gamma(\tau, l) \\ 0 & \text{otherwise.} \end{cases}$$

in the expected test loss for an arbitrary domain $\tau$.

**Theorem D.12.** *Consider the projected linear regression model defined in Section D.2. For any model size $N$, data size $D$, any mixture $h \in \Delta^{K-1}$, and any domain $\tau \in [K]$, when the model is trained with mixture $h$, the bias term in the expected loss for domain $\tau$ in Theorem D.8 has*

$$Bias = \frac{\Gamma\left(1 - \frac{1}{\alpha_\tau}\right)}{\alpha_\tau (2\eta_0)^{1 - \frac{1}{\alpha_\tau}}} \frac{1}{(Dh_\tau)^{1 - \frac{1}{\alpha_\tau}}} + \mathcal{E},$$

*where the error $\mathcal{E}$ is bounded by*

$$|\mathcal{E}| \leq \frac{C_9}{D} + \frac{C_{10}}{N^{\alpha_{\min}\left(1 - \frac{1}{\alpha_\tau}\right)}},$$

*where $C_9$ and $C_{10}$ are constants that only depend on $\alpha$ and $\underline{h}$, but not $h, N, D$.*

*Proof.* We decompose the bias term for domain $\tau$ into two components: the shared head ($k \leq H$) and the disjoint tail ($H < k \leq x_\tau^*$). By definition, the eigenvalue for a tail component of domain $\tau$ in the mixture covariance is $\lambda_k^{(\tau)} = h_\tau k^{-\alpha_\tau}$. Thus, we can write:

$$\text{Bias} = \sum_{k=1}^{H} k^{-\alpha_\tau} \exp\left(-2D\eta_0 \sum_{j=1}^{K} h_j \lambda_k^{(j)}\right) + \sum_{k=H+1}^{x_\tau^*} k^{-\alpha_\tau} \exp\left(-2D\eta_0 h_\tau k^{-\alpha_\tau}\right). \tag{13}$$

**Step 1: Simplify Notation and Bound the Shared Head**
For the shared head ($k \leq H$), the mixture eigenvalue is bounded below by $H^{-\alpha_{\max}}$. Let $A$ denote the upper bound for this head term:

$$A := \sum_{k=1}^{H} k^{-\alpha_\tau} \exp(-2D\eta_0 H^{-\alpha_{\max}}) \leq H \exp(-2D\eta_0 H^{-\alpha_{\max}}).$$

To simplify the tail term, we introduce the constant $c := 2h_\tau D\eta_0$. Let $B$ represent the disjoint tail summation:

$$B := \sum_{k=H+1}^{x_\tau^*} k^{-\alpha_\tau} \exp(-ck^{-\alpha_\tau}).$$

It trivially follows that $B \leq \text{Bias} \leq A + B$, meaning the gap is bounded by $|\text{Bias} - B| \leq A$.

**Step 2: Convert the Discrete Sum to a Continuous Integral**
We approximate $B$ using the continuous function $f(x) := x^{-\alpha_\tau} \exp(-cx^{-\alpha_\tau})$. Taking the derivative $f'(x)$, we find that $f(x)$ increases, peaks, and then decreases, achieving its absolute maximum at $x_0 = c^{\frac{1}{\alpha_\tau}}$. The approximation error between the discrete sum and the continuous integral is strictly bounded by the total variation of $f(x)$ across the interval:

$$\left| B - \int_H^{x_\tau^*} f(x)\mathrm{d}x \right| \leq \int_H^{x_\tau^*} |f'(x)|\mathrm{d}x \leq 2f(x_0) = \frac{2}{c \cdot e} = \frac{1}{eh_\tau D\eta_0}.$$

**Step 3: Evaluate the Continuous Integral**

Applying the change of variables $p = cx^{-\alpha_\tau}$, we have $\mathrm{d}p = -\alpha_\tau cx^{-\alpha_\tau-1}\mathrm{d}x$. The integral transforms into a lower incomplete Gamma function:

$$\int_H^{x_\tau^*} f(x)\mathrm{d}x = \frac{1}{\alpha_\tau c^{1-\frac{1}{\alpha_\tau}}} \int_{p_{\min}}^{p_{\max}} p^{-\frac{1}{\alpha_\tau}} \exp(-p)\mathrm{d}p,$$

where the integration limits are $p_{\min} = c(x_\tau^*)^{-\alpha_\tau}$ and $p_{\max} = cH^{-\alpha_\tau}$. This integral can be evaluated as the complete Gamma function minus the two truncation tails:

$$\int_{p_{\min}}^{p_{\max}} p^{-\frac{1}{\alpha_\tau}} \exp(-p)\mathrm{d}p = \Gamma\left(1 - \frac{1}{\alpha_\tau}\right) - \underbrace{\int_0^{p_{\min}} p^{-\frac{1}{\alpha_\tau}} \exp(-p)\mathrm{d}p}_{\text{Lower Tail}} - \underbrace{\int_{p_{\max}}^\infty p^{-\frac{1}{\alpha_\tau}} \exp(-p)\mathrm{d}p}_{\text{Upper Tail}}.$$

**Step 4: Bound the Truncation Tails**

For the lower tail, since $\exp(-p) \leq 1$:

$$\int_0^{p_{\min}} p^{-\frac{1}{\alpha_\tau}} \exp(-p)\mathrm{d}p \leq \int_0^{p_{\min}} p^{-\frac{1}{\alpha_\tau}} \mathrm{d}p = \frac{\alpha_\tau}{\alpha_\tau - 1} p_{\min}^{1-\frac{1}{\alpha_\tau}} = \frac{\alpha_\tau}{\alpha_\tau - 1} \left(\frac{c}{(x_\tau^*)^{\alpha_\tau}}\right)^{1-\frac{1}{\alpha_\tau}}.$$

For the upper tail, since $p_{\max} > 1$, we have $p^{-\frac{1}{\alpha_\tau}} \leq p_{\max}^{-\frac{1}{\alpha_\tau}}$ for all $p \geq p_{\max}$:

$$\int_{p_{\max}}^\infty p^{-\frac{1}{\alpha_\tau}} \exp(-p)\mathrm{d}p \leq p_{\max}^{-\frac{1}{\alpha_\tau}} \int_{p_{\max}}^\infty \exp(-p)\mathrm{d}p = p_{\max}^{-\frac{1}{\alpha_\tau}} \exp(-p_{\max}) = \frac{H}{c^{\frac{1}{\alpha_\tau}}} \exp\left(-\frac{c}{H^{\alpha_\tau}}\right).$$

**Step 5: Combine Errors to Bound $\mathcal{E}$**

Let $M$ be the principal order (the main term) derived from the complete Gamma function:

$$M := \frac{\Gamma\left(1 - \frac{1}{\alpha_\tau}\right)}{\alpha_\tau c^{1-\frac{1}{\alpha_\tau}}} = \frac{\Gamma\left(1 - \frac{1}{\alpha_\tau}\right)}{\alpha_\tau (2\eta_0)^{1-\frac{1}{\alpha_\tau}}} \frac{1}{(Dh_\tau)^{1-\frac{1}{\alpha_\tau}}}.$$

The total error $\mathcal{E} = \mathrm{Bias} - M$ is bounded by the sum of all accumulated discrepancies: the head error $(A)$, the discrete-to-continuous gap, and the integral truncation tails (multiplied by the prefactor $\frac{1}{\alpha_\tau c^{1-1/\alpha_\tau}}$).

$$|\mathcal{E}| \leq A + \left|B - \int f(x)\mathrm{d}x\right| + (\text{Lower Tail Error}) + (\text{Upper Tail Error})$$

$$\leq H\exp(-2D\eta_0 H^{-\alpha_{\max}}) + \frac{1}{eh_\tau D\eta_0} + \frac{1}{\alpha_\tau - 1} \frac{1}{(x_\tau^*)^{\alpha_\tau - 1}} + \frac{H}{\alpha_\tau c} \exp\left(-\frac{c}{H^{\alpha_\tau}}\right).$$

By applying the inequality $\exp(x) \geq ex$ to the exponential terms and utilizing $x_\tau^* = \Omega(N^{\frac{\alpha_{\min}}{\alpha_\tau}})$ from Lemma D.10, these error terms are bounded asymptotically. Gathering the constants into $C_9$ and $C_{10}$ (which depend strictly on $\alpha$ and the mixture lower bound $\underline{h}$), we obtain the final explicit bound:

$$|\mathcal{E}| \leq \frac{C_9}{D} + \frac{C_{10}}{N^{\alpha_{\min}\left(1-\frac{1}{\alpha_\tau}\right)}}.$$

$\square$

## D.7. The Variance Term

In this section, we analyze the variance terms $\mathrm{Var}_1, \mathrm{Var}_2$ for an arbitrary domain $\tau$. We provide a bound as follows.

**Theorem D.13.** *Consider the projected linear regression model defined in Section D.2. For any $N, D$, any mixture $h \in \Delta^{K-1}, h_i \geq \underline{h}$, and for an arbitrary domain $\tau$, the terms $\mathrm{Var}_1$, $\mathrm{Var}_2$ in Theorem D.8 satisfy*

$$\mathrm{Var}_1 = \left(\sum_{i \in [K]} h_i \sigma_i^2\right) C_{\alpha_\tau} I_\gamma \eta_0^{1/\alpha_\tau} (Dh_\tau)^{1/\alpha_\tau - 1} + \mathcal{E}_1$$

$$\mathrm{Var}_1 \leq \mathrm{Var}_2 \leq \mathrm{Var}_1 + \mathrm{Var}_1 + \frac{C_{14}}{D^{1-\frac{1}{\alpha_\tau}+\varepsilon}}.$$

*where*

$$|\mathcal{E}_1| \le C_1(Dh_\tau)^{-2/3} + C_2DN^{1-2\alpha_\tau},$$

*and $C_{\alpha_\tau}, I_\gamma, C_{14}$ are constants that depend only on $\eta_0, \alpha, H$ but not on mixture $h$.*

### D.7.1. PROOF OF THEOREM D.13

Recall that

$$\mathrm{Var}_1 := \sum_{r_k \le N} \left( \sum_{i \in [K]} h_i \sigma_i^2 \right) a_k \lambda_k \int_0^{D\eta_0} \exp(-2\lambda_k(D\eta_0 - t))\gamma(t)\mathrm{d}t.$$

$$\mathrm{Var}_2 := \sum_{r_k \le N} a_k \lambda_k \int_0^{D\eta_0} \exp(-2\lambda_k(D\eta_0 - t))\gamma(t) \left( \sigma^2 + \frac{C_0}{\underline{h}} \mathbb{E}[\mathbf{w}(t)^\top \mathbf{H} \mathbf{w}(t)] \right) \mathrm{d}t.$$

We start with analyzing $\gamma(t)$. As we defined in Lemma D.7, $\gamma(t) := \eta(\tau^{-1}(t))$. In the following lemma, we derive the exact closed form of $\gamma(\cdot)$.

**Lemma D.14.** *Let $G : \mathbb{R} \to \mathbb{R}$ be the inverse of the map $y \mapsto y + \sin y$. We have*

$$\gamma(t) = \eta_0 \left( 1 + \cos G \left( \frac{t\pi}{D\eta_0} \right) \right).$$

*Proof.* Since $\eta(x) = \eta_0 \left( 1 + \cos \left( \frac{\pi k}{D} \right) \right)$, we have

$$\tau(x) = \int_0^x \eta(a)\mathrm{d}a$$
$$= \eta_0 \frac{D}{\pi} \left( \frac{\pi}{D}x + \sin \left( \frac{\pi}{D}x \right) \right).$$

When $\tau(x) = t$, we have

$$x = \frac{D}{\pi} G \left( \frac{t\pi}{D\eta_0} \right).$$

Therefore

$$\gamma(t) = \eta_0 \left( 1 + \cos G \left( \frac{t\pi}{D\eta_0} \right) \right).$$

$\square$

We first analyze the common term $\sum_{k=1}^N hk^{-2\alpha} \int_0^{D\eta_0} \exp\left(-2hk^{-\alpha}(D\eta_0 - t)\right) \gamma(t)\sigma^2\mathrm{d}t$ that is shared by $\mathrm{Var}_1$ and $\mathrm{Var}_2$.

**Lemma D.15.** *Let $h, \alpha, \eta_0, \sigma > 0$ be constants such that $1 < \alpha < 2$. For an integer $N \in \mathbb{N}^+$ and a continuous variable $D > 0$, consider the sum:*

$$S_1 = \sum_{k=1}^N hk^{-2\alpha} \int_0^{D\eta_0} \exp\left(-2hk^{-\alpha}(D\eta_0 - t)\right) \gamma(t)\sigma^2\mathrm{d}t,$$

*where $G : \mathbb{R} \to \mathbb{R}$ is the inverse function of $y \mapsto y + \sin y$, and $\gamma(t) = \eta_0 \left( 1 + \cos G \left( \frac{t\pi}{D\eta_0} \right) \right)$. Then we have*

$$S_1 = \sigma^2 C_\alpha I_\gamma \eta_0^{1/\alpha}(Dh)^{1/\alpha-1} + \mathcal{E},$$

*where $C_\alpha = \frac{2^{1/\alpha-2}}{\alpha} \Gamma(2 - 1/\alpha)$ and $I_\gamma = \int_0^1 [1 + \cos G(\pi(1-v))] v^{1/\alpha-2}\mathrm{d}v$ are finite constants. The approximation error $\mathcal{E}$ is bounded by*

$$|\mathcal{E}| \le C_6(Dh)^{-2/3} + C_2DN^{1-2\alpha},$$

*where $C_6, C_2$ only depend on $\alpha, \sigma, \eta_0$.*

*Proof.* Let $T = D\eta_0$ and let $s = T - t$. Define the function

$$f(x) := hx^{-2\alpha} \int_0^T \exp\left(-2hx^{-\alpha}s\right) \gamma(T-s)\sigma^2 ds.$$

The target summation can be written as $S_1 = \sum_{k=1}^N f(k)$.

First, we need to bound the schedule function $\gamma(T-s)$ near $s = 0$. By definition, $\gamma(T-s) = \eta_0(1 - \cos x_s)$ where $x_s - \sin x_s = \frac{\pi s}{T}$. For $x_s \in [0, \pi]$, the function $(x_s - \sin x_s)/x_s^3$ achieves its minimum $1/\pi^2$ at $x_s = \pi$. Thus, $x_s - \sin x_s \geq x_s^3/\pi^2$, which implies $x_s \leq \pi(s/T)^{1/3}$. Utilizing $1 - \cos x_s \leq x_s^2/2$, we obtain a strict global bound:

$$\gamma(T-s) \leq \eta_0 \frac{\pi^2}{2} \left(\frac{s}{T}\right)^{2/3}. \tag{14}$$

Now we bound the sum-to-integral gap $|S_1 - \int_1^N f(x)dx|$ by $|S_1 - \int_1^N f(x)dx| \leq \int_1^N |f'(x)|dx$. This inequality follows from analyzing the error on each subinterval $[k, k+1]$. Specifically, the difference can be expressed as $\int_k^{k+1}(f(k)-f(x))dx$. Since $|f(k) - f(x)| \leq \int_k^{k+1} |f'(t)|dt$ for any $x \in [k, k+1]$, summing these local bounds from $k = 1$ to $N - 1$ gives the desired result.

Now we take the derivative of $f(x)$ with respect to $x$:

$$|f'(x)| = \int_0^T hx^{-2\alpha-1} \exp\left(-2hx^{-\alpha}s\right) \left(2\alpha + 2\alpha hsx^{-\alpha}\right) \gamma(T-s)\sigma^2 ds.$$

Therefore, we have

$$\int_1^N |f'(x)|dx = \int_1^N \int_0^T hx^{-2\alpha-1} \exp\left(-2hx^{-\alpha}s\right) \left(2\alpha + 2\alpha hsx^{-\alpha}\right) \gamma(T-s)\sigma^2 dsdx$$

$$= \int_0^T \gamma(T-s)\sigma^2 \int_1^N hx^{-2\alpha-1} \exp\left(-2hx^{-\alpha}s\right) \left(2\alpha + 2\alpha hsx^{-\alpha}\right) dxds$$

$$\leq \int_0^T \gamma(T-s)\sigma^2 \int_1^\infty \left(2\alpha hx^{-2\alpha-1} \exp(-2hx^{-\alpha}s) + 2\alpha sh^2 x^{-3\alpha-1} \exp(-2hx^{-\alpha}s)\right) dxds.$$

Applying the change of variable $v = x^{-\alpha}$ gives

$$\int_1^\infty 2\alpha hx^{-2\alpha-1} \exp(-2hx^{-\alpha}s)dx = 2h \int_0^1 ve^{-2hsv}dv \leq \min\left(h, \frac{1}{2hs^2}\right),$$

and similarly the second term yields $\min\left(\frac{2}{3}h^2 s, \frac{1}{2hs^2}\right)$. Thus, the total variation is bounded by:

$$\int_1^N |f'(x)|dx \leq \sigma^2 \int_0^T \gamma(T-s) \left[\min\left(h, \frac{1}{2hs^2}\right) + \min\left(\frac{2}{3}h^2 s, \frac{1}{2hs^2}\right)\right] ds.$$

Splitting the integral at $s = 1/h$ and substituting the bound (14), we get:

$$\int_1^N |f'(x)|dx \leq \sigma^2 \int_0^{1/h} \eta_0 \frac{\pi^2}{2} \left(\frac{s}{T}\right)^{2/3} \left(\frac{5}{3}h\right) ds + \sigma^2 \int_{1/h}^T \eta_0 \frac{\pi^2}{2} \left(\frac{s}{T}\right)^{2/3} \frac{1}{hs^2} ds \tag{15}$$

$$= \frac{5\pi^2}{6}\sigma^2 \eta_0 T^{-2/3} h \left[\frac{3}{5}\left(\frac{1}{h}\right)^{5/3}\right] + \frac{\pi^2}{2}\sigma^2 \eta_0 T^{-2/3} \frac{1}{h} \left[3\left(\frac{1}{h}\right)^{-1/3}\right] \tag{16}$$

$$= 2\pi^2 \sigma^2 \eta_0 (hT)^{-2/3} = 2\pi^2 \sigma^2 \eta_0^{1/3} (Dh)^{-2/3}. \tag{17}$$

Next, we evaluate the continuous continuous integral $\int_1^N f(x)dx$. Since integrate over $(1, N)$ requires incomplete gamma function, we split $\int_1^N$ into $\int_0^\infty - \int_0^1 - \int_N^\infty$. We first integrate over $x \in (0, \infty)$ and extract the main order $M := \int_0^\infty f(x)dx$.

Swapping the integration order yields:

$$\int_0^\infty f(x)\mathrm{d}x = \sigma^2 \int_0^T \gamma(T-s)\left(\int_0^\infty hx^{-2\alpha}\exp(-2hx^{-\alpha}s)\mathrm{d}x\right)\mathrm{d}s$$

$$= \sigma^2 C_\alpha h^{1/\alpha-1}\int_0^T \gamma(T-s)s^{1/\alpha-2}\mathrm{d}s.$$

Using the dimensionless variable $v = s/T$, we factor out $T$ to match the stated integral $I_\gamma$:

$$\int_0^\infty f(x)\mathrm{d}x = \sigma^2 C_\alpha I_\gamma \eta_0^{1/\alpha}(Dh)^{1/\alpha-1} =: M.$$

Let $\mathcal{E} := |S_1 - M|$, we decompose the error as $\mathcal{E} = \left(S_1 - \int_1^N f(x)\mathrm{d}x\right) - \int_0^1 f(x)\mathrm{d}x - \int_N^\infty f(x)\mathrm{d}x$. We bound the upper and lower truncation tails individually. For the upper integral tail $x \in [0,1]$, we substitute (14) into $f(x)$ and evaluate the $s$-integral exactly:

$$f(x) \leq \sigma^2 \int_0^\infty hx^{-2\alpha}\exp(-2hx^{-\alpha}s)\left[\eta_0 \frac{\pi^2}{2}\left(\frac{s}{T}\right)^{2/3}\right]\mathrm{d}s$$

$$= \frac{\pi^2}{2}\sigma^2 \eta_0 hx^{-2\alpha}T^{-2/3}\Gamma(5/3)(2hx^{-\alpha})^{-5/3} = \frac{\pi^2\Gamma(5/3)}{2^{8/3}}\sigma^2 \eta_0 T^{-2/3}h^{-2/3}x^{-\alpha/3}.$$

Here, $\Gamma(\cdot)$ denotes the Gamma function, defined as

$$\Gamma(x) = \int_0^\infty t^{x-1}e^{-t}\,dt$$

for $x > 0$ (or $\mathrm{Re}(x) > 0$). Note that this is distinct from the previously defined mapping function $\Gamma(\cdot,\cdot)$, which takes two arguments. Integrating this bounding function over $x \in [0,1]$ directly gives:

$$\int_0^1 f(x)\mathrm{d}x \leq \frac{\pi^2\Gamma(5/3)}{2^{8/3}}\sigma^2 \eta_0(hT)^{-2/3}\int_0^1 x^{-\alpha/3}\mathrm{d}x = \frac{3\pi^2\Gamma(5/3)}{2^{8/3}(3-\alpha)}\sigma^2 \eta_0^{1/3}(Dh)^{-2/3}. \tag{18}$$

For the lower integral tail $x \in [N,\infty)$, we use the trivial bound $\gamma(T-s) \leq 2\eta_0$:

$$\int_N^\infty f(x)\mathrm{d}x \leq \sigma^2 \int_N^\infty\left(\int_0^T hx^{-2\alpha}\exp(-2hx^{-\alpha}s)(2\eta_0)\mathrm{d}s\right)\mathrm{d}x \tag{19}$$

$$\leq \sigma^2 \int_N^\infty hx^{-2\alpha}(2\eta_0 T)\mathrm{d}x = \frac{2\eta_0\sigma^2 hT}{2\alpha-1}N^{1-2\alpha} = \frac{2\eta_0^2\sigma^2 Dh}{2\alpha-1}N^{1-2\alpha}. \tag{20}$$

Combining all error sources yields

$$\mathcal{E} \leq 2\pi^2\sigma^2 \eta_0^{1/3}(Dh)^{-2/3} + \frac{3\pi^2\Gamma(5/3)}{2^{8/3}(3-\alpha)}\sigma^2 \eta_0^{1/3}(Dh)^{-2/3} + \frac{2\eta_0^2\sigma^2 Dh}{2\alpha-1}N^{1-2\alpha}$$

$$\leq C_6(Dh)^{-2/3} + C_2 DN^{1-2\alpha},$$

where $C_6, C_2$ only depends on $\alpha, \sigma, \eta_0$. $\qquad\square$

Now we focus on $\mathrm{Var}_1, \mathrm{Var}_2$. Note that $\mathrm{Var}_1$ and $\mathrm{Var}_2$ only differs in $\mathbb{E}[\mathbf{w}^\top(t)\mathbf{H}\mathbf{w}(t)]$ and $\mathbb{E}[\mathbf{w}^\top(t)\mathbf{H}\mathbf{w}(t)] + \sigma^2$ is exactly the expected training loss. In the following discussion, we first show that the expected training loss satisfies the following theorem.

**Lemma D.16.**

$$\mathbb{E}\left[\mathbf{w}^\top(t)\mathbf{H}\mathbf{w}(t)\right] \leq \min\left(\frac{C'}{t^{1-\frac{1}{\alpha_{\max}}}} + \frac{C''}{N^{\alpha_{\min}\left(1-\frac{1}{\alpha_{\max}}\right)}} + \frac{C'''}{D^{1-\frac{1}{\alpha_{\min}}}}, L\right),$$

*where $L$ is defined in Assumption D.5 and $C', C'', C'''$ are constants that only depend on $\alpha, H$.*

*Proof.* Since $\mathbf{H} = \sum h_i \mathbf{A}_i$, we have

$$\mathbf{w}^\top(t)\mathbf{H}\mathbf{w}(t) = \sum_{i=1}^{K} h_i \mathbf{w}^\top(t)\mathbf{A}_i \mathbf{w}(t).$$

Now we focus on a certain domain $j$. By Theorem D.8, we have

$$\mathbb{E}[\mathbf{w}^\top(t)\mathbf{A}_j\mathbf{w}(t)] \leq \underbrace{\sum_{k:\, r_k > N} a_k}_{\text{Approx}}$$

$$+ \underbrace{\sum_{k:\, r_k \leq N} \exp(-2\lambda_k t)a_k}_{\text{Bias}}$$

$$+ \underbrace{\sum_{k:\, r_k \leq N} a_k \lambda_k \int_0^t \exp(-2\lambda_k(D\eta_0 - s))\gamma(t)\left(\sum_{i\in[K]} h_i\sigma_i^2 + \frac{C_0}{\underline{h}}\mathbb{E}\left[\mathbf{w}(s)^\top \mathbf{H}\mathbf{w}(s)\right]\right) \mathrm{d}t}_{\text{Var}_2},$$

where $a_k = u_j^\top \mathbf{A}_\tau u_k$. Directly applying Theorem D.9 and Lemma D.10, we have

$$\sum_{k:\, r_k > N} a_k \leq \frac{(x_j^*(h))^{1-\alpha_j}}{\alpha_j - 1} + \frac{C_{13}}{N^{\alpha_{\min}}}$$

$$\leq \mathcal{O}\left(\frac{1}{N^{\frac{\alpha_j - 1}{\alpha_j}\alpha_{\min}}}\right)$$

where $\mathcal{O}$ does not hide any term depends on $h$.

By Theorem D.12, we have

$$\sum_{k:\, r_k \leq N} \exp(-2\lambda_k D\eta_0)a_k \leq \frac{\Gamma\left(1 - \frac{1}{\alpha_j}\right)}{\alpha_j(2\eta_0)^{1-\frac{1}{\alpha_j}}}\frac{1}{(h_j t)^{1-\frac{1}{\alpha_j}}} + \frac{C_9}{t} + \frac{C_{10}}{N^{\alpha_{\min}\left(1-\frac{1}{\alpha_j}\right)}}$$

Let $\sigma'^2 := \sum_{i\in[K]} h_i\sigma_i^2 + \frac{C_0}{\underline{h}}L$, the last term can be upper bound as:

$$\sum_{k:\, r_k \leq N} a_k \lambda_k \int_0^t \exp(-2\lambda_k(D\eta_0 - s))\gamma(t)\left(\sum_{i\in[K]} h_i\sigma_i^2 + \frac{C_0}{\underline{h}}\mathbb{E}\left[\mathbf{w}(s)^\top \mathbf{H}\mathbf{w}(s)\right]\right)\mathrm{d}t$$

$$\leq \sum_{k:\, r_k \leq N} a_k \lambda_k \int_0^{D\eta_0} \exp(-2\lambda_k(D\eta_0 - s))\gamma(t)\sigma'^2 \mathrm{d}t.$$

$$\leq \sum_{k\in[K]} h_j k^{-2\alpha_j} \int_0^{D\eta_0} \exp(-2hk^{-\alpha_j}(D\eta_0 - s))\gamma(t)\sigma'^2 \mathrm{d}t.$$

By Lemma D.15, we have

$$\sum_{k\in[K]} h_j k^{-2\alpha_j} \int_0^{D\eta_0} \exp(-2hk^{-\alpha_j}(D\eta_0 - s))\gamma(t)\sigma'^2 \mathrm{d}t$$

$$\leq \sigma'^2 C_\alpha I_\gamma \eta_0^{1/\alpha_j}(Dh_j)^{1/\alpha_j - 1} + C_6(Dh_j)^{-2/3}.$$

Combining these three terms, we have

$$\mathbb{E}[\mathbf{w}^\top(t)\mathbf{A}_j\mathbf{w}(t)] \leq \mathcal{O}\left(\frac{1}{N^{\frac{\alpha_j-1}{\alpha_j}\alpha_{\min}}}\right) + \frac{\Gamma\left(1-\frac{1}{\alpha_j}\right)}{\alpha_j(2\eta_0)^{1-\frac{1}{\alpha_j}}} \frac{1}{(h_jt)^{1-\frac{1}{\alpha_j}}} + \frac{C_9}{t} + \frac{C_{10}}{N^{\alpha_{\min}\left(1-\frac{1}{\alpha_j}\right)}}$$
$$+ \sigma'^2 C_\alpha I_\gamma \eta_0^{1/\alpha_j}(Dh_j)^{1/\alpha_j-1} + C_6(Dh_j)^{-2/3}.$$

Since $\mathbb{E}\left[\sum_{i=1}^{K} h_i\mathbf{w}^\top(t)\mathbf{A}_i\mathbf{w}(t)\right] \leq \max_{j\in[K]}\mathbb{E}[\mathbf{w}^\top(t)\mathbf{A}_j\mathbf{w}(t)]$ and $t < D\eta_0 \leq N^{\frac{\alpha_{\min}}{1+\varepsilon}}\eta_0$, extracting minimum exponents on $t, N, D$, we have

$$\mathbb{E}\left[\sum_{i=1}^{K} h_i\mathbf{w}^\top(t)\mathbf{A}_i\mathbf{w}(t)\right] \leq \frac{C'}{t^{1-\frac{1}{\alpha_{\max}}}} + \frac{C''}{N^{\alpha_{\min}\left(1-\frac{1}{\alpha_{\max}}\right)}} + \frac{C'''}{D^{1-\frac{1}{\alpha_{\min}}}},$$

where $C', C'', C'''$ are constants only depend on $\alpha, H$. With Assumption D.5, we complete the proof. $\qquad\square$

After we obtain a rough bound on $\mathbf{w}^\top(t)\mathbf{H}\mathbf{w}(t)$, we are ready to get a refined bound on $\text{Var}_2$. The next Lemma analyze the crucial term in $\text{Var}_2 - \text{Var}_1$.

**Lemma D.17.** *Let $h, \alpha, \alpha_2, \eta_0, C > 0$ be positive constants. For an integer $N \in \mathbb{N}^+$, and a continuous scale parameter $D > 0$, consider the sum:*

$$S_2 = \sum_{k=1}^{N} hk^{-2\alpha} \int_0^{D\eta_0} \exp\left(-2hk^{-\alpha}(D\eta_0 - t)\right)\gamma(t)\min(C, t^{-\alpha_2})dt,$$

*where $G: \mathbb{R} \to \mathbb{R}$ is the inverse function of $y \mapsto y + \sin y$, and $\gamma(t) = \eta_0\left(1 + \cos G\left(\frac{t\pi}{D\eta_0}\right)\right)$.*

*The summation $S_2$ is strictly bounded by:*

$$S_2 \leq C_3(Dh)^{-2/3} + C_4 DN^{1-2\alpha} + C_5(Dh)^{1/\alpha-1}D^{-1},$$

*where $C_5$ only depends on $\alpha, C$ which are all constant.*

*Proof.* Let $T = D\eta_0$. Applying the change of variable $s = T - t$, we define the function

$$f(x) := hx^{-2\alpha}\int_0^T \exp\left(-2hx^{-\alpha}s\right)\gamma(T-s)\min(C, (T-s)^{-\alpha_2})ds.$$

The target summation can be exactly written as $S_2 = \sum_{k=1}^{N} f(k)$.

We define the continuous principal integral as $\mathcal{I} := \int_0^\infty f(x)dx$ (integrating from 0 to infinity yields simpler results) and

$$\mathcal{E} := \left(\sum_{k=1}^{N} f(k) - \int_1^N f(x)dx\right) - \int_0^1 f(x)dx - \int_N^\infty f(x)dx. \tag{21}$$

To bound $\left|\sum_{k=1}^{N} f(k) - \int_1^N f(x)dx\right|$, we first take the derivative of $f(x)$ with respect to $x$:

$$f'(x) = \int_0^T hx^{-2\alpha-1}\exp\left(-2hx^{-\alpha}s\right)\left(-2\alpha + 2\alpha hsx^{-\alpha}\right)\gamma(T-s)\min(C, (T-s)^{-\alpha_2})ds.$$

Therefore,

$$|f'(x)| \leq \int_0^T hx^{-2\alpha-1}\exp\left(-2hx^{-\alpha}s\right)\left|-2\alpha + 2\alpha hsx^{-\alpha}\right|\gamma(T-s)\min(C, (T-s)^{-\alpha_2})ds$$
$$\leq \int_0^T hx^{-2\alpha-1}\exp\left(-2hx^{-\alpha}s\right)\left(2\alpha + 2\alpha hsx^{-\alpha}\right)\gamma(T-s)Cds.$$

By Equation 21, we obtain:

$$|\mathcal{E}| \leq \int_1^N |f'(x)|\mathrm{d}x + \int_0^1 f(x)\mathrm{d}x + \int_N^\infty f(x)\mathrm{d}x$$

$$\leq 2\pi^2 C\eta_0^{1/3}(Dh)^{-2/3} + \frac{3\pi^2\Gamma(5/3)}{2^{8/3}(3-\alpha)}C\eta_0^{1/3}(Dh)^{-2/3} + \frac{2\eta_0^2 Ch}{2\alpha-1}DN^{1-2\alpha}$$

$$= C_3(Dh)^{-2/3} + C_4 DN^{1-2\alpha}.$$

The second inequality follows the same steps as in Equations (15), (18) and (19).

Next, we evaluate $\mathcal{I}$. Swapping the order of integration yields:

$$\mathcal{I} = \int_0^\infty f(x)\mathrm{d}x = \int_0^T \gamma(T-s)\min(C,(T-s)^{-\alpha_2})\left(\int_0^\infty hx^{-2\alpha}\exp\left(-2hx^{-\alpha}s\right)\mathrm{d}x\right)\mathrm{d}s.$$

Applying the substitution $v = x^{-\alpha}$ with $\mathrm{d}x = -\frac{1}{\alpha}v^{-1/\alpha-1}\mathrm{d}v$, the inner integral evaluates to $C_\alpha h^{1/\alpha-1}s^{1/\alpha-2}$, where $C_\alpha = \frac{2^{1/\alpha-2}}{\alpha}\Gamma(2-1/\alpha)$. Reversing the initial temporal substitution ($t = T - s$), the principal integral becomes exactly:

$$\mathcal{I} = C_\alpha h^{1/\alpha-1}\int_0^T (T-t)^{1/\alpha-2}\gamma(t)\min(C,t^{-\alpha_2})\mathrm{d}t.$$

Since $\min(C,t^{-\alpha_2})$ is a constant when $t$ is small, we split the integration domain at $t = T/2$ such that

$$\mathcal{I} = \underbrace{C_\alpha h^{1/\alpha-1}\int_0^{\frac{T}{2}} (T-t)^{1/\alpha-2}\gamma(t)\min(C,t^{-\alpha_2})\mathrm{d}t}_{\mathcal{I}_1} + \underbrace{C_\alpha h^{1/\alpha-1}\int_{\frac{T}{2}}^T (T-t)^{1/\alpha-2}\gamma(t)\min(C,t^{-\alpha_2})\mathrm{d}t}_{\mathcal{I}_2}.$$

For the first half-domain $t \in [0,T/2]$, the term $(T-t) \geq T/2$. Since $1/\alpha - 2 < 0$, we have $(T-t)^{1/\alpha-2} \leq (T/2)^{1/\alpha-2}$. Furthermore, $\gamma(t) \leq 2\eta_0$. We bound the integral of the cutoff function over this local domain by its global integral over $(0,\infty)$:

$$\int_0^{T/2} \min(C,t^{-\alpha_2})\mathrm{d}t \leq \int_0^{C^{-1/\alpha_2}} C\mathrm{d}t + \int_{C^{-1/\alpha_2}}^\infty t^{-\alpha_2}\mathrm{d}t$$

$$= C^{1-1/\alpha_2} + \frac{1}{\alpha_2-1}C^{1-1/\alpha_2}$$

$$= \frac{\alpha_2}{\alpha_2-1}C^{1-1/\alpha_2}.$$

Multiplying these individual maximum bounds, we obtain:

$$\mathcal{I}_1 \leq C_\alpha h^{1/\alpha-1}\left(\frac{T}{2}\right)^{1/\alpha-2}(2\eta_0)\left(\frac{\alpha_2}{\alpha_2-1}C^{1-1/\alpha_2}\right)$$

$$= \frac{2^{3-1/\alpha}\alpha_2 C_\alpha \eta_0^{1/\alpha-1}}{\alpha_2-1}C^{1-1/\alpha_2}\frac{(Dh)^{1/\alpha-1}}{D}.$$

For the second half-domain $t \in [T/2,T]$, the variable is bounded away from zero. Consequently, the minimum function is strictly bounded by its algebraic tail: $\min(C,t^{-\alpha_2}) \leq t^{-\alpha_2} \leq (T/2)^{-\alpha_2} = 2^{\alpha_2}T^{-\alpha_2}$. Factoring out this upper bound allows us to conservatively extend the remaining integral to the full domain $[0,T]$:

$$\mathcal{I}_2 \leq 2^{\alpha_2}T^{-\alpha_2}C_\alpha h^{1/\alpha-1}\int_{T/2}^T (T-t)^{1/\alpha-2}\gamma(t)\mathrm{d}t$$

$$\leq 2^{\alpha_2}T^{-\alpha_2}C_\alpha h^{1/\alpha-1}\int_0^T (T-t)^{1/\alpha-2}\gamma(t)\mathrm{d}t.$$

Applying the dimensionless change of variable $v = 1 - t/T$ sets $\mathrm{d}t = -T\mathrm{d}v$ and $T - t = Tv$, fully recovering the constant $I_\gamma$ from Lemma D.15:

$$\int_0^T (T-t)^{1/\alpha-2}\gamma(t)\mathrm{d}t = T^{1/\alpha-1}\eta_0 \int_0^1 v^{1/\alpha-2}\left[1 + \cos G(\pi(1-v))\right]\mathrm{d}v$$
$$= T^{1/\alpha-1}\eta_0 I_\gamma.$$

Substituting this back and recalling $T = D\eta_0$, we obtain:

$$\mathcal{I}_2 \le 2^{\alpha_2}(D\eta_0)^{-\alpha_2}C_\alpha h^{1/\alpha-1}\left((D\eta_0)^{1/\alpha-1}\eta_0 I_\gamma\right)$$
$$= 2^{\alpha_2}C_\alpha I_\gamma \eta_0^{1/\alpha-\alpha_2}\frac{(Dh)^{1/\alpha-1}}{D^{\alpha_2}}.$$

Summing the evaluated components gives the final strict bound:

$$S_2 \le \mathcal{I}_1 + \mathcal{I}_2 + |\mathcal{E}|$$
$$\le C_3(Dh)^{-2/3} + C_4 DN^{1-2\alpha} + \frac{2^{3-1/\alpha}\alpha_2 C_\alpha \eta_0^{1/\alpha-1}}{\alpha_2 - 1}C^{1-1/\alpha_2}\frac{(Dh)^{1/\alpha-1}}{D} + 2^{\alpha_2}C_\alpha I_\gamma \eta_0^{1/\alpha-\alpha_2}\frac{(Dh)^{1/\alpha-1}}{D^{\alpha_2}}$$
$$\le C_3(Dh)^{-2/3} + C_4 DN^{1-2\alpha} + C_5(Dh)^{1/\alpha-1}D^{-1},$$

where $C_3, C_4, C_5$ are absolute constants that only depend on $\alpha, \alpha_2, C, \eta_0$, completing the proof. $\qquad\square$

Now, we are ready to proof Theorem D.13.

Let

$$\mathcal{E}_1 := \mathrm{Var}_1 - \left(\sum_{i\in[K]} h_i\sigma_i^2\right)C_{\alpha_\tau}I_\gamma \eta_0^{1/\alpha_\tau}(Dh_\tau)^{1/\alpha_\tau-1},$$

where $C_{\alpha_\tau}$ and $I_\gamma$ are the constants defined in Lemma D.15. Then by Lemma D.15, $\mathcal{E}_1$ satisfies

$$|\mathcal{E}_1| \le C_6(Dh_\tau)^{-2/3} + C_2 DN^{1-2\alpha_\tau} + \int_0^{D\eta_0} \exp(-2H^{-\alpha_{\max}}(D\eta_0 - t))\gamma(t)\sigma'^2\mathrm{d}t$$
$$\le C_6(Dh_\tau)^{-2/3} + C_2 DN^{1-2\alpha_\tau} + \frac{\pi^2}{2}(D\eta_0)^{-2/3}\Gamma(5/3)(2h_\tau H^{-\alpha_{\max}})^{-5/3}$$
$$\le C_1(Dh_\tau)^{-2/3} + C_2 DN^{1-2\alpha_\tau}.$$

$\int_0^{D\eta_0} \exp(-2H^{-\alpha_{\max}}(D\eta_0 - t))\gamma(t)\sigma'^2\mathrm{d}t$ is induced by the error of the shared head and $C_1 = C_6 + \frac{\pi^2}{2}\eta_0^{-2/3}\Gamma(5/3)H^{5\alpha/3}$ which does not depend on $h$.

By Lemma D.16, we have

$$\mathrm{Var}_1 \le \mathrm{Var}_2 \le \mathrm{Var}_1 + \sum a_k\lambda_k \int_0^{D\eta_0} \exp(-2\lambda_k(D\eta_0 - t))\gamma(t)\frac{C_0}{h}\left(\min\left(\frac{C'}{1-\frac{1}{\alpha_{\max}}}, L\right) + \frac{C''}{N^{\alpha_{\min}\left(1-\frac{1}{\alpha_{\max}}\right)}} + \frac{C'''}{D^{1-\frac{1}{\alpha_{\min}}}}\right)\mathrm{d}t$$

By Lemma D.17, we have

$$\sum a_k\lambda_k \int_0^{D\eta_0} \exp(-2\lambda_k(D\eta_0 - t))\gamma(t)\frac{C_0}{h}\left(\min\left(\frac{C'}{1-\frac{1}{\alpha_{\max}}}, L\right) + \frac{C''}{N^{\alpha_{\min}\left(1-\frac{1}{\alpha_{\max}}\right)}} + \frac{C'''}{D^{1-\frac{1}{\alpha_{\min}}}}\right)\mathrm{d}t$$
$$\le (C_1 + C_3)(Dh_\tau)^{-2/3} + C_4 DN^{1-2\alpha_\tau} + C_5(Dh)^{1/\alpha_\tau-1}D^{-1}$$
$$+ \left(\left(\sum_{i\in[K]} h_i\sigma_i^2\right)C_{\alpha_\tau}I_\gamma \eta_0^{1/\alpha_\tau}(Dh_\tau)^{1/\alpha_\tau-1} + |\mathcal{E}_1|\right)\cdot\left(\frac{C''}{N^{\alpha_{\min}\left(1-\frac{1}{\alpha_{\max}}\right)}} + \frac{C'''}{D^{1-\frac{1}{\alpha_{\min}}}}\right).$$

Therefore,

$$\mathrm{Var}_2 \leq \mathrm{Var}_1 + (C_1 + C_3)(Dh_\tau)^{-2/3} + C_4 DN^{1-2\alpha_\tau} + C_5(Dh)^{1/\alpha_\tau - 1}D^{-1}$$

$$+ \left( \left( \sum_{i \in [K]} h_i \sigma_i^2 \right) C_{\alpha_\tau} I_\gamma \eta_0^{1/\alpha_\tau} (Dh_\tau)^{1/\alpha_\tau - 1} + |\mathcal{E}_1| \right) \cdot \left( \frac{C''}{N^{\alpha_{\min}\left(1 - \frac{1}{\alpha_{\max}}\right)}} + \frac{C'''}{D^{1-\frac{1}{\alpha_{\min}}}} \right)$$

$$\leq \mathrm{Var}_1 + \frac{C_{14}}{D^{1-\frac{1}{\alpha_\tau}+\varepsilon}}.$$

The last in equality is by finding the minimum exponents on $D$ and thus $C_{14}$ is independent on $h$.

## D.8. Proof of the Main Theorem

Combining Theorem D.9, Theorem D.12 and Theorem D.13, we will have our final main theorem, with the following minor assumption.

**Assumption D.18.** We assume that the value of the term $\frac{\Gamma\left(1-\frac{1}{\alpha_\tau}\right)}{\alpha_\tau(2\eta_0)^{1-\frac{1}{\alpha_\tau}}}$ is much larger than $\left( \sum_{i \in [K]} h_i \sigma_i^2 \right) C_{\alpha_\tau} I_\gamma \eta_0^{1/\alpha_\tau}$ for all $h \in \Delta(K-1)$, so that

$$A(h) := \frac{\Gamma\left(1-\frac{1}{\alpha_\tau}\right)}{\alpha_\tau(2\eta_0)^{1-\frac{1}{\alpha_\tau}}} + \left( \sum_{i \in [K]} h_i \sigma_i^2 \right) C_{\alpha_\tau} I_\gamma \eta_0^{1/\alpha_\tau} \approx \text{a constant } A.$$

Note that when $\alpha \in [1, 3], \eta \in [0.1, 1]$ and $\sigma_i^2 \leq 1$, we have $A(h) \in [8, 9]$.

Then our main theorem can be formally stated as follows.

**Theorem D.19** (Formal Statement of Theorem 4.1). *Consider the projected linear regression model defined in Section D.2 and consider an arbitrary domain $\tau \in [K]$. Let $x^*$ be the solution to Problem 1 with parameters $c_i = \frac{1}{\alpha_i - 1}, b_i = \alpha_i - 1$. Then for any $\epsilon > 0$, there exist $N_0$ and $D_0$ such that for all $N > N_0$, $D > D_0$, and $h$ satisfying Assumption D.1, the expected test loss on domain $\tau$ is bounded by:*

$$L_\tau(h, N, D) \geq c_\tau(x^*(N, h)_\tau)^{-b_\tau}(1 - \epsilon) + \frac{A(h)}{(Dh_\tau)^{1-\frac{1}{\alpha_\tau}}}(1 - \epsilon) + \sigma_i^2,$$

$$L_\tau(h, N, D) \leq c_\tau(x^*(N, h)_\tau)^{-b_\tau}(1 + \epsilon) + \frac{A(h)}{(Dh_\tau)^{1-\frac{1}{\alpha_\tau}}}(1 + \epsilon) + \sigma_i^2,$$

*where $A(h) := \frac{\Gamma\left(1-\frac{1}{\alpha_\tau}\right)}{\alpha_\tau(2\eta_0)^{1-\frac{1}{\alpha_\tau}}} + \left( \sum_{i \in [K]} h_i \sigma_i^2 \right) C_{\alpha_\tau} I_\gamma \eta_0^{1/\alpha_\tau}$. With Theorem D.18, we have $A(h) \approx$ a constant A, and thus*

$$L_\tau(h, N, D) \geq c_\tau(x^*(N, h)_\tau)^{-b_\tau}(1 - \epsilon) + \frac{A}{(Dh_\tau)^{1-\frac{1}{\alpha_\tau}}}(1 - \epsilon) + \sigma_i^2,$$

$$L_\tau(h, N, D) \leq c_\tau(x^*(N, h)_\tau)^{-b_\tau}(1 + \epsilon) + \frac{A}{(Dh_\tau)^{1-\frac{1}{\alpha_\tau}}}(1 + \epsilon) + \sigma_i^2,$$

*for a constant A.*

*Proof.* Applying Theorem D.9 bounds the relative approximation error for all $h$ satisfying Assumption D.1:

$$\frac{\left| \mathrm{Approx} - \frac{(x^\tau)^{1-\alpha_\tau}}{\alpha_\tau - 1} \right|}{\frac{(x^\tau)^{1-\alpha_\tau}}{\alpha_\tau - 1}} \leq \frac{\frac{\alpha_\tau}{N^{\alpha_\tau+1}}}{\frac{(x^{*\tau})^{1-\alpha_\tau}}{\alpha_\tau - 1}} \leq \frac{\alpha_\tau(\alpha_\tau - 1)}{N^{\alpha_\tau+1-\frac{\alpha_{\min}}{\alpha_\tau}(\alpha_\tau-1)}} \leq \frac{\alpha_\tau(\alpha_\tau - 1)}{N^{\alpha_\tau-\alpha_{\min}+1+\frac{\alpha_{\min}}{\alpha_\tau}}}.$$

Next, by Theorem D.12, for all $h$ satisfying Assumption D.1, the relative bias is bounded by:

$$\frac{\left| \text{Bias} - \frac{\Gamma\left(1-\frac{1}{\alpha_\tau}\right)}{\alpha_\tau(2\eta_0)^{1-\frac{1}{\alpha_\tau}}} \frac{1}{(Dh_\tau)^{1-\frac{1}{\alpha_\tau}}} \right|}{\frac{\Gamma\left(1-\frac{1}{\alpha_\tau}\right)}{\alpha_\tau(2\eta_0)^{1-\frac{1}{\alpha_\tau}}} \frac{1}{(Dh_\tau)^{1-\frac{1}{\alpha_\tau}}}} \leq \frac{\frac{C_9}{D} + \frac{C_{10}}{N^{\alpha_{\min}\left(1-\frac{1}{\alpha_\tau}\right)}}}{\frac{\Gamma\left(1-\frac{1}{\alpha_\tau}\right)}{\alpha_\tau(2\eta_0)^{1-\frac{1}{\alpha_\tau}}} \frac{1}{(Dh_\tau)^{1-\frac{1}{\alpha_\tau}}}}$$

$$\leq C_{11} D^{-1/\alpha_\tau} + C_{12} D^{-\varepsilon(1-1/\alpha_\tau)},$$

where we define $C_{11} := C_9 \cdot \left( \frac{\Gamma\left(1-\frac{1}{\alpha_\tau}\right)}{\alpha_\tau(2\eta_0)^{1-\frac{1}{\alpha_\tau}}} \right)^{-1}$ which is independent of $h$.

Similarly, by Theorem D.13, for all $h$ satisfying Assumption D.1, the relative variance satisfies:

$$\frac{\left| \text{Var} - \sigma^2 C_{\alpha_\tau} I_\gamma \eta_0^{1/\alpha_\tau} (Dh_\tau)^{1-1/\alpha_\tau} \right|}{\sigma^2 C_{\alpha_\tau} I_\gamma \eta_0^{1/\alpha_\tau} (Dh_\tau)^{1-1/\alpha_\tau}} \leq \frac{(C_3 + 2C_1 + C_5)(Dh_\tau)^{-2/3} + (C_4 + C_2)DN^{1-2\alpha_\tau}}{\sigma^2 C_{\alpha_\tau} I_\gamma \eta_0^{1/\alpha_\tau} (Dh_\tau)^{1-1/\alpha_\tau}}$$

$$\leq C_7 D^{-1/\alpha_\tau+1/3} + C_8 D^{-2(1+\varepsilon)(1-1/\alpha_\tau)},$$

where $C_7$ and $C_8$ are constants independent of $h$ and $\epsilon$.

Note that the exponents of $N$ and $D$ in the denominators are strictly positive; that is, $\alpha_\tau - \alpha_{\min} + 1 + \frac{\alpha_{\min}}{\alpha_\tau} > 0$ and $1/3 - 1/\alpha_\tau > 0$. Therefore, we can choose sufficiently large $D_0$ and $N_0$ such that for all $h$ the error bound is small enough:

$$\frac{2\alpha_\tau(\alpha_\tau - 1)}{N_0^{\alpha_\tau-\alpha_{\min}+1+\frac{\alpha_{\min}}{\alpha_\tau}}} \leq \epsilon,$$

$$C_{11} D_0^{-1/\alpha_\tau} + C_{12} D_0^{-\varepsilon} \leq \epsilon,$$

$$C_7 D_0^{-1/\alpha_\tau+1/3} + C_8 D_0^{-\varepsilon} \leq \epsilon.$$

Consequently, for all $N > N_0$ and $D > D_0$ satisfying Assumption D.6, for all $h$ satisfying Assumption D.1, the relative errors are uniformly bounded by $\epsilon$. This implies that the components fall within the following intervals:

$$\text{Approx} \in \left[ \frac{(x_\tau^*)^{1-\alpha_\tau}}{\alpha_\tau - 1}(1-\epsilon), \frac{(x_\tau^*)^{1-\alpha_\tau}}{\alpha_\tau - 1}(1+\epsilon) \right],$$

$$\text{Bias} \in \left[ \frac{\Gamma\left(1-\frac{1}{\alpha_\tau}\right)}{\alpha_\tau(2\eta_0)^{1-\frac{1}{\alpha_\tau}}} \frac{1}{(Dh_\tau)^{1-\frac{1}{\alpha_\tau}}}(1-\epsilon), \frac{\Gamma\left(1-\frac{1}{\alpha_\tau}\right)}{\alpha_\tau(2\eta_0)^{1-\frac{1}{\alpha_\tau}}} \frac{1}{(Dh_\tau)^{1-\frac{1}{\alpha_\tau}}}(1+\epsilon) \right],$$

$$\text{Var} \in \left[ \sigma^2 C_{\alpha_\tau} I_\gamma \eta_0^{1/\alpha_\tau} (Dh_\tau)^{1-1/\alpha_\tau}(1-\epsilon), \sigma^2 C_{\alpha_\tau} I_\gamma \eta_0^{1/\alpha_\tau} (Dh_\tau)^{1-1/\alpha_\tau}(1+\epsilon) \right].$$

Finally, by Theorem D.8, adding these three components together completes the proof. $\square$

## E. Proof of Proposition 4.2

The objective function for the bi-level optimization is given by $\mathcal{J}(h) = \sum_{i=1}^K w_i[A_i(Dh_i)^{-a_i} + c_i(x_i^*(h))^{-b_i}]$. To find the gradient with respect to a mixture weight $h_k$, we apply the chain rule, noting that the data term depends explicitly on $h_k$ while the capacity term depends implicitly on $h$ through the optimal allocation $x^*(h)$:

$$\frac{d\mathcal{J}}{dh_k} = -w_k a_k A_k D^{-a_k} h_k^{-a_k-1} + \sum_{i=1}^K w_i \frac{\partial L_i^{\text{cap}}}{\partial x_i} \frac{\partial x_i^*(h)}{\partial h_k}. \tag{22}$$

The first term is obtained by direct differentiation. To evaluate the second term (the capacity gradient), we invoke the KKT conditions of the inner optimization problem $\min_x \sum h_i c_i x_i^{-b_i}$ subject to $\sum x_i = N$. The stationarity condition implies $\lambda = h_i b_i c_i x_i^{-(b_i+1)}$, which simplifies the marginal capacity loss to $\frac{\partial L_i^{\text{cap}}}{\partial x_i} = -b_i c_i x_i^{-b_i-1} = -\frac{\lambda}{h_i}$.

Next, we derive the Jacobian $\frac{\partial x_i}{\partial h_k}$ using implicit differentiation. Taking the logarithm of the stationarity condition yields $\ln \lambda = \ln h_i + \text{const} - (b_i + 1) \ln x_i$. Differentiating both sides with respect to $h_k$ gives:

$$\frac{1}{\lambda} \frac{\partial \lambda}{\partial h_k} = \frac{\delta_{ik}}{h_i} - \frac{b_i + 1}{x_i} \frac{\partial x_i}{\partial h_k}. \tag{23}$$

Letting $\gamma_i = \frac{x_i}{b_i+1}$, we can express the sensitivity of the capacity allocation as $\frac{\partial x_i}{\partial h_k} = \gamma_i \left( \frac{\delta_{ik}}{h_i} - \frac{1}{\lambda} \frac{\partial \lambda}{\partial h_k} \right)$. We solve for the unknown multiplier derivative by differentiating the capacity constraint $\sum_j x_j = N$ with respect to $h_k$, which implies $\sum_j \frac{\partial x_j}{\partial h_k} = 0$. Summing the sensitivity equation over all $j$ yields:

$$\sum_{j=1}^{K} \gamma_j \left( \frac{\delta_{jk}}{h_j} - \frac{1}{\lambda} \frac{\partial \lambda}{\partial h_k} \right) = 0 \quad \implies \quad \frac{1}{\lambda} \frac{\partial \lambda}{\partial h_k} = \frac{\gamma_k}{Z h_k}, \tag{24}$$

where $Z = \sum_{j=1}^{K} \gamma_j$. Substituting this back, we obtain the Jacobian:

$$\frac{\partial x_i}{\partial h_k} = \gamma_i \left( \frac{\delta_{ik}}{h_i} - \frac{\gamma_k}{Z h_k} \right). \tag{25}$$

Finally, we substitute the marginal loss $-\frac{\lambda}{h_i}$ and the Jacobian back into the chain rule equation for the capacity term:

$$\begin{aligned}
\frac{\partial \mathcal{J}_{\text{cap}}}{\partial h_k} &= \sum_{i=1}^{K} w_i \left( -\frac{\lambda}{h_i} \right) \gamma_i \left( \frac{\delta_{ik}}{h_i} - \frac{\gamma_k}{Z h_k} \right) \\
&= -\lambda \left( \frac{w_k \gamma_k}{h_k^2} - \frac{\gamma_k}{Z h_k} \sum_{i=1}^{K} \frac{w_i \gamma_i}{h_i} \right).
\end{aligned} \tag{26}$$

We define the weighted baseline $\bar{R} = \frac{1}{Z} \sum_{i=1}^{K} \frac{w_i \gamma_i}{h_i}$. Factoring out the common terms and substituting $\gamma_k = \frac{x_k^*}{b_k+1}$, the capacity gradient simplifies to $\frac{\lambda x_k^*}{h_k(b_k+1)} \left( \bar{R} - \frac{w_k}{h_k} \right)$. combining this with the data gradient completes the proof. $\qquad \square$

