# OpenReview forum: "Explaining Data Mixing Scaling Laws"
_ICML.cc/2026/Conference — ICML 2026 regular_

### Official Review · Reviewer_AGKe · 2026-03-03

**Soundness:** 3
**Presentation:** 3
**Significance:** 3
**Originality:** 3
**Overall Recommendation:** 4
**Confidence:** 2

**Summary:**

This paper studies data mixture selection for multi-domain pretraining, where training data are sampled from $K$ domains according to mixture weights $h$, and the goal is to find an optimal $h^*$ that minimizes loss on a target distribution $w$ under a fixed budget. Since existing empirical data mixing laws are largely black-box curve fits with limited theoretical explanation, the authors propose a unified theoretical framework that extends single-domain scaling-law theories to the multi-domain setting under a “Shared Head, Disjoint Tail” assumption. The framework identifies two mechanisms governing domain loss: capacity competition and noise reduction. Concretely, they derive an interpretable loss form where the expected domain loss decomposes into a capacity-allocation term plus a domain-local noise term. They then cast optimal-mixture search as a bi-level optimization and solve it with Online Mirror Descent. Empirically, the proposed models achieve the lowest/second-lowest fitting error while using fewer parameters than heuristic baselines. In additional experiments training 200M and 700M models on 4 Pile domains, the mixture predicted by their method yields the lowest validation loss among compared approaches.

**Compliance With Llm Reviewing Policy:**

Affirmed.

**Final Justification:**

The rebuttal was convincing and addressed my concerns well. That said, because I am not fully confident in the depth of my own expertise in this area, I prefer not to revise my score upward.

**Key Questions For Authors:**

Please refer to the weaknesses listed above.

**Limitations:**

yes

**Strengths And Weaknesses:**

**Strengths**

1. **Unified, theory-driven framework beyond black-box fitting:** The paper clearly motivates that existing data-mixing scaling laws are largely empirical curve fits, and proposes a principled framework by extending scaling-law theories (e.g., Quantization / projected linear regression views) to the multi-domain setting.
2. **Interpretable decomposition of mixture effects:** It provides a clean explanation of why domains interact under mixing by decomposing domain loss into capacity competition (coupled across domains due to finite model capacity) and a data-amount-driven noise term, and leverages this structure for both loss prediction and optimal mixture search.
3. **Empirical validation on fitting and mixture optimization:** The proposed models achieve low fitting error on mixture loss landscape, and the predicted optimal mixtures yield the lowest validation loss in additional 4-domain training experiments at 200M and 700M scale.

**Weaknesses**
1. **Strong modeling assumptions (Shared Head / Disjoint Tail):** The analysis relies on an idealized assumption that tails are effectively disjoint across domains; the paper acknowledges this as an approximation. Robustness under substantial cross-domain tail overlap (or stronger domain correlations) remains unclear.
2. **Practicality and generalization scope:** (i) Parameter fitting depends on non-convex optimization (potentially sensitive to initialization / computationally heavy), and (ii) the work mainly targets predicting losses within the training-domain mixture; extensions to unseen domains or downstream tasks are left as future work.

---

> ### Author Rebuttal · Authors · 2026-03-31
>
> We thank the reviewer for valuable feedback and constructive comments.
>
> > 1. Strong modeling assumptions (Shared Head / Disjoint Tail): The analysis relies on an idealized assumption that tails are effectively disjoint across domains; the paper acknowledges this as an approximation. Robustness under substantial cross-domain tail overlap (or stronger domain correlations) remains unclear.
>
> We highlight that our disjoint-tail model remains a good approximation even when domain tails overlap significantly.
> - To empirically validate the robustness of our model, we conducted a synthetic experiment simulating $K=7$ domains with a $40%$ global tail overlap. When fitting our disjoint-tail model to this highly overlapping data, the MRE was remarkably low at just 0.07%. This confirms that even under substantial cross-domain tail overlap, our model remains a good approximation.
> - Furthermore, our real-world experiments on the Pile dataset naturally contain domains with tail overlap: GitHub and StackExchange heavily share programming syntax and computational concepts; the Pile-CC (Common Crawl) domain acts as a massive "global" overlap, intersecting with other domains in the corpus. Despite this overlap, our model fits the loss landscape with the lowest MRE.
> - We provide an explanation for the robustness of our model as follows. Compared to the disjoint skills, the status of overlapping skills (learned vs unlearned) will be more stable as the mixture changes, which means they are more likely to take up a constant fraction of the "effective capacity", and the loss induced by overlapping skills will behave more like a constant. Therefore, our models remains a good approximation when overlapping skills exist.
>
> > 2. Practicality and generalization scope: (i) Parameter fitting depends on non-convex optimization (potentially sensitive to initialization / computationally heavy), and (ii) the work mainly targets predicting losses within the training-domain mixture; extensions to unseen domains or downstream tasks are left as future work.
>
> - While the optimization is indeed non-convex, we found it to be computationally manageable in practice. By employing the basin-hopping algorithm, our fitting process reliably converges within 10 iterations. We acknowledge that achieving this rapid convergence relies on appropriate hyperparameter tuning—specifically, adjusting the temperature and step size to effectively navigate and escape local minima. However, once these standard hyperparameters are set, the computational overhead of this fitting process is negligible compared to the massive computational cost of the actual LLM pre-training it aims to guide.

---

> > ### Author Rebuttal · Reviewer_AGKe · 2026-04-01
> >
> > Thank you for the careful and sincere clarification. I find the response convincing. That said, because I am not fully confident in the depth of my own expertise in this area, I would prefer not to revise my score upward.

---

> > > ### Author Response · Authors · 2026-04-05
> > >
> > > We thank the reviewer for their careful evaluation of our work and for reading our responses. Please let us know if any further questions arise.

---

### Official Review · Reviewer_n3Zm · 2026-03-11

**Soundness:** 2
**Presentation:** 2
**Significance:** 2
**Originality:** 2
**Overall Recommendation:** 3
**Confidence:** 2

**Summary:**

The authors propose a theoretical model for explaining the data mixtures in neural scaling laws. They recognize capacity allocation and noise reduction as major factors that determine the model's performance on target distribution when trained using data mixtures from varied domains. Additionally, they predict an optimal weighting scheme using their theoretically backed scaling law to obtain the best downstream performance.

**Compliance With Llm Reviewing Policy:**

Affirmed.

**Final Justification:**

I thank the authors for the rebuttal and the reply. After reading the other reviews (specifically the questions from Reviewer j8aK) and the rebuttal/reply, I would like to maintain my current score.

**Key Questions For Authors:**

It is known that dynamic data mixture strategies, where h changes over the course of training, consistently outperform static fixed data mixtures. It would be great if the authors shed some light on how their theory would be used or transferred to a dynamic data mixture strategy?

Please also see Weaknesses above.

**Limitations:**

Yes

**Strengths And Weaknesses:**

The paper introduces bi-level optimization to find optimal weights. This explains why the optimal training mixture differs from the target distribution. Empirically, it is known that h* is not equal to w, but this work gives it a theoretical explanation.

The paper connects neural scaling laws and the transfer learning intuitions into a unified framework which makes it quite valuable looking at drawn insights. The capacity competition and the statistical transfer loss working in opposite directions simultaneously seem to be be well captured by Theorem 3.1.

The paper is well written and easy to read.

Weaknesses:

I am not convinced by the shared teacher assumption in the linear regressor model. Different domains may have genuinely conflicting optimal parameters. The shared teacher forces all domain differences to be in the same covariance structure.  So, the theory structurally cannot predict when you should exclude a domain from training entirely. Since every domain shares the same teacher, adding any domain data can only help or be neutral. But it is empirically well known that some data sources do degrade performance and therefore should be excluded from training. Please correct me if I am wrong.

The power law exponent, alpha, is set to be constant across k. That implies that the decay rate is the same for easy skills and hard skills. But in practice, the spectrum tends to decay faster for common skills than the long-tailed/rare ones. It would be great if an ablation study/discussion would be available regarding this in the paper.

---

> ### Author Rebuttal · Authors · 2026-03-31
>
> We thank the reviewer for valuable feedback and constructive comments.
>
> > 1. I am not convinced by the shared teacher assumption in the linear regressor model. Different domains may have genuinely conflicting optimal parameters...
> - **Importantly, our framework supports the existence of genuinely conflicting optimal parameters. The projection matrix $S$ in our model inherently induces the parameter conflicts commonly observed in  practice.**
> - The teacher model $\theta^\*$ does not represent the model being trained. Rather, it represents all underlying relevant parameters and can be infinite-dimensional. And the feature vector $x$ represents all underlying relevant features and may also be infinite-dimensional. Within this teacher model, we assume no parameter conflict, as we can always construct a higher-dimensional parameter $\theta^\*$ to resolve any conflicts. For example, given two domains with $\theta^\*_1 \neq \theta^*_2$, we can define $\theta^\* = (\theta^\*_1, \theta^\*_2)^{\top}$, representing data from domain 1 as $(x_1, 0)^{\top}$ and data from domain 2 as $(0, x_2)^{\top}$.
> - In practice, the model and data sizes are finite. The data collection process is modeled by a projection matrix $S$, which projects the infinite $x$ into a lower-dimensional observable space $\tilde{x} = Sx$. Crucially, it is this projection matrix $S$ that induces the parameter conflicts commonly observed in the real world.
> - **Example:** Consider ${\theta^{\*}}^{\top} = [ \theta_0^\top , \theta_1^\top, \dots, \theta_k^\top]$
> where $\theta_0 \in R^d$ represents shared knowledge, and each $\theta_i \in R^d$ represents the specialized ground truth for domain $i$. When we sample a data point from domain $i$, its original high-dimensional feature is $x = (x_{\text{head}}^{\top}, \mathbf{0}^{\top}, \ldots, x_{i}^{\top}, \ldots, \mathbf{0}^{\top})^{\top}$.  We assume a data collection process that extracts only the non-zero features and the student model only observes the projected features $\tilde{x} = \mathbf{S}x = (x_{\text{head}}^{\top}, x_{i}^{\top})^{\top}$ and learns a parameter $\tilde\theta = [\theta_{\text{head}}^{\top}, \theta_{\text{tail}}^{\top}]^{\top}$. This creates parameter conflicts: the student is forced to use the single parameter block $\tilde{\theta}_{\text{tail}}$ to predict all the specialized tails. For a synthetic experiment, please see https://anonymous.4open.science/r/ICML2026_response-5FEB/ICML_rebuttal_reviewer3.pdf.
>
> > The power law exponent, alpha, is set to be constant across k...But in practice, the spectrum tends to decay faster for common skills than the long-tailed/rare ones..
>   - We thank the reviewer for this insightful observation.
>   - To clarify, the differing decay rates between common and long-tail skills do not compromise our framework. We assume common skills are always learned, incurring negligible loss. **The domain-wise power-law exponents in our model are designed to only characterize the distribution of the long-tail, specialized skills**, which is where the competition for finite model capacity occurs.
>   - In addition, our use of a constant $\alpha$ follows the standard macroscopic approximation in scaling law literature. At the massive scale of LLM pretraining, attempting to model fine-grained rates across the entire skill spectrum is intractable. Treating $\alpha$ as a constant abstracts away microscopic variations to provide a tractable functional form that, as our experiments demonstrate, accurately predicts macroscopic loss behaviors. We also evaluated our law on the 7-domain SlimPajama data mixtures recently released by Shukor et al. (2025) (made available after the ICML submission deadline). Please see our response to Weakness 2 by reviewer 6QL9.
>
> > It is known that dynamic data mixture strategies outperform static fixed data mixtures. It would be great if the authors shed some light on how their theory would be used or transferred to a dynamic data mixture strategy?
>
> - We thank the reviewer for the insightful question.
> - A key finding of our work is that model capacity plays a critical role in determining the optimal data mixture. Specifically, domains compete not only for the training data budget, but also for the model's finite capacity. Since current dynamic strategies mostly focus on data allocation without considering model capacity competition, our findings might provide a new direction for improving these methods.
> - In addition, to our knowledge, dynamic data mixture strategies are not widely deployed in practice due to their computational overhead. Currently, the performance improvements often do not justify the increased training costs (for further context on this matter, please see the "Discussions" section in (Kang et al., 2025, "Autoscale: Scale-aware Data Mixing for Pre-training LLMs"). To advance these strategies, we believe gaining a deeper theoretical understanding is crucial, which is also a primary motivation for this work.

---

> > ### Author Rebuttal · Reviewer_n3Zm · 2026-04-02
> >
> > Thank you very much for the rebuttal. After reading the other reviews and the rebuttal, I am convinced with empricial results, but still not with theoretical justification. So I would prefer to keep my current rating.

---

> > > ### Author Response · Authors · 2026-04-05
> > >
> > > We thank the reviewer for reviewing our responses and providing prompt feedback. Although the latest response does not specify which aspect of the theoretical justification remains unconvincing, leaving us to infer the exact point of disagreement, we understand that our model is mathematically dense, which may lead to lingering concerns about how our framework captures genuinely conflicting parameters across domains. To address this, we further clarify our previous response.
> > > - **Conflict Emerges via Projection:** First, as we explained in our previous response, our current framework is able to model genuinely conflicting parameters across domains. To clarify, we distinguish between two settings within our framework:
> > >     - **The Underlying Data Generation Process (No Conflict):** The teacher model $\theta^\*$, which can be infinite-dimensional, represents all underlying relevant parameters, operating on all relevant underlying features $x$ to generate a label $y$. Because a model with infinite capacity can leverage all relevant features without making trade-offs, more data always helps. Thus, it is perfectly reasonable that $\theta^\*$ itself has no parameter conflict.
> > >     - **The Practical Training Environment (Conflict Emerges):** In practice, we train a finite-size model on finite-size data. This is represented by the student model $\hat{\theta}$, which is trained on observed, finite-dimensional features $\tilde{x} = Sx$, where the projection matrix $S$ represents the data collection process. The optimal $\hat{\theta}$ can differ for data from different domains, as we explained in the previous response.
> > > - **Generality of the Phenomenon:** Second, the concrete example provided in the previous response is not an uncommon edge case. Whenever the dimension of the observed data is lower than the underlying data dimension, parameter conflict naturally emerges: the matrix $S$ naturally induces parameter conflict.
> > >
> > > - **Empirical Validation:** Finally, from an empirical perspective, our framework already predicts the loss landscape with the lowest MRE and finds the best data mixture. While extending the framework to explicitly define different $\theta_i^*$ for different domains could be explored in future work, our current formulation already mathematically allows for parameter conflict and demonstrates strong empirical validity.
> > >
> > > We remain fully available during this rebuttal period and welcome the reviewer to engage in further discussion regarding any specific concerns they may still have.

---

### Official Review · Reviewer_j8aK · 2026-03-12

**Soundness:** 2
**Presentation:** 2
**Significance:** 3
**Originality:** 3
**Overall Recommendation:** 3
**Confidence:** 5

**Summary:**

This paper proposes a theoretical framework to model data mixing scaling laws based on multi-domain linear regression setting, and gives an algorithm for optimising the training mixing proportions for a task that is a weighted linear combination of multiple domains. The framework uses model capacity allocation to model domain interaction and loss on the domain by the number of skills the model has acquired. They use the theoretical framework to propose an algorithm for optimizing data mixing proportions with less parameters than current sota.

**Compliance With Llm Reviewing Policy:**

Affirmed.

**Key Questions For Authors:**

1. Is it possible to quantify or estimate how much worse off is the optimal training mixture for scaling laws $h^*$ is from the actual optimal training mixture based on the error of scaling law of each domain? Does your framework (when properly defined with explicit error bounds instead of approximate sign) give a bound on the final error found by your procedure and actually optimal error?
2. Why is the extended quantization model relevant given its limitations?
3. What are “skills” for the Pile dataset?
4. Why are the spectral assumptions used in extended linear regression scaling model reasonable to hold when trying to apply the same framework in the text domain? What is the rationale behind even expecting the same framework to work?

**Limitations:**

Yes.

**Strengths And Weaknesses:**

Strengths:
1. Extending the scaling laws in linear regression model to multiple domains is an original and interesting idea. This is particularly important because it gives theoretical backing and clear modelling assumptions for understanding interaction between different domains, which was, as far as I know, missing in existing literature.
2. The bottleneck to identify optimal mixtures in practice is the number of parameters, so the proposed algorithm which achieves good performance with less parameters is a significant contribution. It also validates the applicability of this approach to more complex settings than linear regression.


Weaknesses:
1. Most of the key definitions are unclear and underdefined. For example, definitions of optimal training mixture and coverage threshold are confusing. In line 129, and in Equations (2) and (4), the optimal training mixture $h^*$ is defined as the training mixture that optimizes the loss predicted by the scaling laws ($\sum_i w_i f_i(h, N,D)$). But that is not the optimal training mixture - the optimal mixture minimizes the actual test loss, not a prediction of it. This distinction is important and should be clearly clarified in the paper. The coverage threshold is defined through the effective capacity, which is not defined properly and is interchangeably used with model size and “total mass of skills”. Can these be properly defined? The concept of “skills” is not clearly defined or explain.
2. Main claims of a “theoretical framework” are informal. This to me loses the point of having a theoretical framework if I cannot understand it or formally state it. The approximate sign is used freely throughout the paper without any clarification what it means. The main quantities are not defined properly (see #1). The main theorem in the paper (theorem 3.1) is informal - approximate sign is never defined or quantified what it means. This makes the presentation of this theoretical paper very bad and difficult to follow.
3. The significance of Extended Quantization Model is unclear and a bit confusing. Why should we introduce a model that fails to capture that $h*\neq w$ , which is the most important empirical observation on which the whole area of optimizing data mixtures is based (otherwise we would always just set $h*=w$ and there would be no need to optimize $h*$)?
4. The paper is missing a discussion of some related works, which also consider explaining mechanics of data mixing. (a) Domain-aware scaling laws uncover data synergy by Hamidieh et al.: the paper considers how to model direct domain interaction with scaling laws, and shows how their form of scaling law can explain synergies between certain domains observed in practice (such as code and web data). This, to my knowledge, is not covered by Shukor et al and Ye et al papers, so it would be interesting to see if this framework and Capacity Competition can model these domain synergies. (b) Shift is Good: Mismatched Data Mixing Improves Test Performance by Medvedev et al.: the paper shows how to find optimal training mixtures for a task that is given by a test mixture of K domains, quantifies the optimal training mixture and the error improvement over the baseline, which is exactly the same setup as here (once we scale the weights to sum to 1). They also find closed form solutions for the optimal training mixtures for some forms of losses predicted by fitting laws. It could be interesting to see if any of the results in the current paper are captured by Medvedev et al, and, if yes, to compare how does the prediction of the optimal mixture and its performance from section 4 compares to the predictions from Medvedev et al.

---

> ### Author Rebuttal · Authors · 2026-03-31
>
> We thank the reviewer for valuable feedback and constructive comments.
>
>
> ## Major Clarifications:
>
> > W1: In line 129, and in Equations (2) and (4), .. But that is not the optimal training mixture - the optimal mixture minimizes the actual test loss, not a prediction of it.
> - The notation $h^\*$ **is never intended to be a definition for the ground truth optimal mixture.** It simply defines how the optimal mixture is computed in practice. In practice, people first fit the loss with tractable functional forms, and then compute an optimal mixture $h^\*$ with these fitted functions. Similarly, in Section 3, $h^\*$ is the computed optimal mixture when the domain losses are predicted by our model. We do not claim $h^\*$ equals the ground truth optimal mixture, nor do we theoretically bound the error of this optimal mixture. The theoretical model is mainly used to derive an approximate functional form, and its effectiveness is validated by experiments. We will clarify this.
>
> > Q1: Is it possible to quantify how much worse off is the optimal training mixture is ...
>
> - We thank the reviewer for the interesting theoretical question. While possible, we do not attempt to bound the optimality gap for two fundamental reasons:
>     1. **Limits of Transferability:** While prior literature suggests that macroscopic functional forms derived from simplified linear models could explain scaling laws for large-scale models, there is no evidence that theoretical error bounds share this transferability: there is no theoretical or empirical basis to assume that the highly non-linear transformer will adhere to strict bounds derived for simple linear models.
>     2. **Computational Infeasibility:** Even if we were to derive such a bound, empirically verifying it at the LLM scale is a computationally impossible task.
>
>      Our primary objective is practical utility. So **instead of an unverifiable theoretical bound, our methodology relies on empirical validation**: our model yields lower MRE and discovers better mixtures than existing heuristic baselines. Please also see our **New Experiments with Post-Submission Data Release** in response to Reviewer 6QL9.
>
> > W2: ... The approximate sign is used freely throughout the paper without any clarification what it means.
>
> - While we simplified notation in the main text, our framework follows the formal model from prior work (Lin et al., 2024; Li et al., 2025), and all theoretical results can be formally stated. Our use of $\approx$ denotes asymptotic bounds: $\text{Loss} \approx f(\cdot)$ formally means $\text{Loss} = \Theta(f(\cdot))$. We will replace this notation and explicitly restate our main theorem in its fully formal version:$$L_i(h, N, D)=\Theta\left((x^*_i)^{-a_i} \right)+\Theta\left(\frac{1}{\left(Dh_i\right)^{b_i}} \right)+ E$$ Here, the right-hand side provides an asymptotic approximation, following (Lin et al., 2024; Li et al., 2025).
>
> > Q4: Why are the spectral assumptions reasonable to hold when applied in the text domain?
> - As discussed in Line 275-288, the spectral assumption is analogous to the power-law skill distributions in the text-domain literature (e.g. Michaud et al., 2023; Pan et al., 2025). These works focus on text data and assume a power-law skill distribution. In particular, Michaud et al. (2023) empirically validated this assumption on the TinyStories dataset. Pan et al. (2025) ground this assumption in Zipf’s Law: word frequencies in natural language follow a power law.
> - The ultimate justification for this assumption is that it produces functional forms that work in practice, as validated in our LLM experiments.
>
> ## Other Important Responses:
>
> > W4:  Related work.
>
> - We thank the reviewer for pointing us to these highly relevant papers. But we want to clarify the following regarding Medvedev et al.:
>   - Their framework assumes strict domain independence—meaning that data from one domain has no impact on the loss of other domains. **This assumption fails to explain the non-trivial domain interaction, which is the main contribution of our work.**
>   - Due to this independence assumption, **adopting their closed-form optimal mixture yields an outcome that is equivalent to the BiMix law (Ge et al., 2024)**, which is outperformed by our model.
>
> > W3 and Q2: The significance of Extended Quantization Model
>
> - It isolates the primary driver of the non-trivial domain interaction: because data from different domains compete for finite model capacity. Crucially, even this simplified model achieves a lower MRE than previous empirical scaling laws.
>
> > W1: The effective capacity, model size, total mass of skills. Q3: What are “skills” for the Pile dataset?
>
> - We will consistently use model size ($N$), removing the other two. We set $N$ = # of model parameters /$10^6$, following (Shukor et al. 2025). We have also tested other values and found that the resulting MREs remain almost identical. For the term "skills", see the definition of Quantum in (Michaud et al. 2023).

---

> > ### Author Rebuttal · Reviewer_j8aK · 2026-04-02
> >
> > > "It simply defines how the optimal mixture is computed in practice. In practice, people first fit the loss with tractable functional forms, and then compute an optimal mixture $h^*$ with these fitted functions."
> >
> > I understand that. But both from practical and theoretical point of view, the optimal mixture for the tractable funcional form loss is not something I care about. I really care about the mixture that is optimal for the true loss.
> >
> > > "We thank the reviewer for the interesting theoretical question. [...]  instead of an unverifiable theoretical bound, our methodology relies on empirical validation"
> >
> > This is not a theoretical question. It's conceivable that such a quantification can be done with a completely empirical method and could give a way to empirically estimate how close we are to the best mixture we can hope for. I feel like the question is being unfairly disregarded. Again, as I said above, this is a very relevant question both from empirical, practical, and theoretical standpoints. If I care about getting the best possible performance but I am ok with being a bit worse than that, it would be very helpful to know what mixture is good enough.
> >
> > > "formally means..."
> >
> > What is $\Theta$ with respect to?
> >
> > > "Their framework assumes strict domain independence"
> >
> > Not in section 6 on transfer learning. Still, it might be worth commenting in the related works section, and it would further clarify the contribution of the paper.
> >
> > > "For the term "skills", see the definition of Quantum in (Michaud et al. 2023)."
> >
> > My point is just that if you choose to use this term, it might be useful to describe it in half a sentence in the paper.

---

> > > ### Author Response · Authors · 2026-04-03
> > >
> > > We thank the reviewer for carefully reading our response and for providing detailed, constructive feedback.
> > > >I understand that. But both from practical and theoretical point of view, the optimal mixture for the tractable functional form loss is not something I care about. I really care about the mixture that is optimal for the true loss.
> > >
> > > - **We completely agree that the ultimate goal is to minimize the true loss, and this is exactly the objective of all LLM data mixing literature and this work. Our optimal mixture yields the lowest true loss compared to previous baselines. This is achieved by proposing a model that better approximates the true loss and the true optimal mixture.**
> > >
> > > >It's conceivable that such a quantification can be done with a completely empirical method and could give a way to empirically estimate how close we are to the best mixture we can hope for.
> > >
> > > ## **Unfortunately, there is currently no feasible method for us to quantify the error in the optimal mixture for LLMs, even through completely empirical methods.**
> > >
> > > - Firstly, as previously discussed, **error analysis from linear models are not guaranteed to transfer to large-scale LLMs**, so running simulation on linear model is pointless.
> > >
> > > - For LLMs, finding the ground truth optimal mixture is highly challenging, **there is currently no computationally feasible method that guarantees to find the ground truth optimal mixture, even for a small 100M model**:
> > >     - The most straightforward empirical approach would be a grid search over the entire space of possible mixtures; however, this is computationally prohibitive. Even for a small-scale model (100M parameters / 1B tokens), training a single mixture takes 1 hour on an A100 machine. If we assume a setting with 4 domains, discretize the mixture weights into integer percentages (which is necessary, as changing the weight by 1% noticeably changes the model loss), this still yields **156849 possible combinations and will take 17.91 years to train on an A100 machine.** For reference, **in the scaling law literature, researchers typically train only tens of mixtures and fit a function to predict the loss landscape.**
> > >     - Due to the prohibitive computational cost, existing approaches generally fall into three categories: (1) heuristic search, (2) greedy adjustment of mixture weights during training, and (3) scaling law methods. **Crucially, none of these approaches guarantees finding the ground-truth optimal mixture.** Please see the Related Work section in the appendix for these references.
> > >
> > > >What is $\Theta$ with respect to?
> > >
> > > - We thank the reviewer for pointing out this ambiguity. In the original equation, $\Theta(\cdot)$ is with respect to the model size $N$ and the data size $D$, and we prove a bound that holds for all mixtures $h$ simultaneously. To clarify this, our main theorem can be revised as follows:
> > > For any $\epsilon>0$, there exist $N_0,D_0$ such that for all $N>N_0,D>D_0$ and for all $h \in \Delta^{K-1}$ satisfying $h_i \geq \underline{h}>0$, the expected domain loss $L_i$ satisfies
> > > $$E_i+(1-\epsilon)\cdot A(Dh_i)^{1/\alpha_i-1}+(1-\epsilon)\frac{{x_i^\*(h,N)}^{1-\alpha_i}}{\alpha_i-1}\le L_i \le E_i+(1+\epsilon)\cdot A(Dh_i)^{1/\alpha_i-1}+(1+\epsilon)\frac{{x_i^\*(h,N)}^{1-\alpha_i}}{\alpha_i-1},$$
> > > where $\underline{h}$ is a small positive constant, $A=\frac{\gamma(1-1/\alpha_i,1)}{\alpha_i(2\eta_0)^{1-1/\alpha_i}},$  and $\gamma$ is lower incomplete gamma function.
> > >
> > > - Basically, we prove that when the model size and the data size are large enough, the approximate functional form will be accurate.  Again, we highlight that we do not aim to bound the error of this approximation. We derive an approximate functional form and verify its validity by experiments.
> > >
> > > > Not in section 6 on transfer learning. Still, it might be worth commenting in the related works section, and it would further clarify the contribution of the paper.
> > > - We thank the reviewer for the further explanation and the suggestion. In Section 6 of the referenced paper, the authors assume an overlapping error term that depends on the total amount of data (there is a typo in their Model 6.1, where the sum should be $n_1 + \cdots + n_K$). When deciding the optimal mixture, the total data size is fixed. This term reduces to a constant and does not introduce domain interaction. We will certainly cite and discuss this work in our revision.
> > >
> > > > My point is just that if you choose to use this term, it might be useful to describe it in half a sentence in the paper.
> > >
> > > - Thanks for the suggestion. We will add more detailed explanations in the revision.
> > >
> > > Please let us know if any questions remain, we welcome any further questions and discussions.

---

### Official Review · Reviewer_6QL9 · 2026-03-12

**Soundness:** 3
**Presentation:** 3
**Significance:** 3
**Originality:** 3
**Overall Recommendation:** 4
**Confidence:** 2

**Summary:**

This paper proposes a unified framework to explain scaling laws for mixed data. The authors first dissect two major theories in the single-domain setting—the Quantization Model and the Linear Regression Model—and then extend them to the multi-domain scenario by introducing the highly generalized theoretical assumption of "shared head, disjoint tail." Subsequently, starting from two key mechanisms, Capacity Competition and Noise Reduction, the authors formalize loss prediction and optimal mixture search via convex and bi-level optimization. In the experimental section, validation on 200M and 700M models demonstrates that this theory outperforms existing empirical formulas in fitting accuracy and can effectively predict the optimal mixture.

**Compliance With Llm Reviewing Policy:**

Affirmed.

**Ethical Review Concerns:**

Final Recommendation and Justification

I maintain my recommendation of Weak Accept. This paper makes a substantial theoretical contribution by proposing a unified framework that extends single-domain scaling laws to multi-domain settings through the “shared head, disjoint tail” assumption and formalizes loss prediction and optimal mixture search via convex and bi-level optimization. My initial assessment highlighted its strengths in originality (the unified framework and mechanistic explanations), significance (providing theory-driven insights beyond empirical fitting), and soundness (rigorous modeling and experimental validation on medium-scale models). However, I noted weaknesses concerning the validation of the idealized disjoint-tail assumption and the limited experimental scope (primarily 200M/700M models).

The authors’ rebuttal has largely addressed these concerns. They provided strong additional empirical support: (1) new experiments on recently released 7-domain SlimPajama data, showing their framework outperforms the current SOTA in fitting accuracy and optimal mixture prediction at the 1B scale; (2) a synthetic experiment demonstrating that the model remains a good approximation even under significant tail overlap (MRE 0.07%); and (3) preliminary extrapolation results scaling from 200M to 700M parameters. These responses directly mitigate the experimental scope weakness and offer empirical validation for the core assumption, thereby strengthening the paper’s overall soundness. The provided figure on mixture deviation (Q2) also enhances clarity.

While the rebuttal reinforces the paper’s empirical grounding and practical utility, the fundamental approximation inherent in the disjoint-tail assumption—despite the authors’ reasonable justification—still places some bound on the framework’s generality. Thus, the work remains a technically solid and insightful contribution that clearly advances the sub-area, but its impact is somewhat tempered by this idealized modeling choice. The authors’ comprehensive responses have reinforced my prior assessment that the paper merits acceptance, and I believe it will provide a valuable foundation for future research on scaling laws for mixed data.

**Final Justification:**

Final Recommendation and Justification

I maintain my recommendation of Weak Accept. This paper makes a substantial theoretical contribution by proposing a unified framework that extends single-domain scaling laws to multi-domain settings through the “shared head, disjoint tail” assumption and formalizes loss prediction and optimal mixture search via convex and bi-level optimization. My initial assessment highlighted its strengths in originality (the unified framework and mechanistic explanations), significance (providing theory-driven insights beyond empirical fitting), and soundness (rigorous modeling and experimental validation on medium-scale models). However, I noted weaknesses concerning the validation of the idealized disjoint-tail assumption and the limited experimental scope (primarily 200M/700M models).

The authors’ rebuttal has largely addressed these concerns. They provided strong additional empirical support: (1) new experiments on recently released 7-domain SlimPajama data, showing their framework outperforms the current SOTA in fitting accuracy and optimal mixture prediction at the 1B scale; (2) a synthetic experiment demonstrating that the model remains a good approximation even under significant tail overlap (MRE 0.07%); and (3) preliminary extrapolation results scaling from 200M to 700M parameters. These responses directly mitigate the experimental scope weakness and offer empirical validation for the core assumption, thereby strengthening the paper’s overall soundness. The provided figure on mixture deviation (Q2) also enhances clarity.

While the rebuttal reinforces the paper’s empirical grounding and practical utility, the fundamental approximation inherent in the disjoint-tail assumption—despite the authors’ reasonable justification—still places some bound on the framework’s generality. Thus, the work remains a technically solid and insightful contribution that clearly advances the sub-area, but its impact is somewhat tempered by this idealized modeling choice. The authors’ comprehensive responses have reinforced my prior assessment that the paper merits acceptance, and I believe it will provide a valuable foundation for future research on scaling laws for mixed data.

**Key Questions For Authors:**

Q1: Under the “Shared Head, Disjoint Tail” assumption, how can the degree of tail overlap be quantified? In your framework, would this correspond to a measurable ‘domain confusion’ metric, and how does it affect the final ‘effective capacity requirement’ function?

Q2: In the bilevel optimization problem, you mention that the optimal mixture is not equal to the target distribution. Could you provide more intuitive examples to illustrate the magnitude and pattern of this deviation?

Q3: How applicable is your method to larger scales (e.g., models with tens of billions of parameters)? Are there any preliminary experiments or theoretical extrapolations?

**Limitations:**

yes

**Strengths And Weaknesses:**

Strengths：

1.Deep and Systematic Theoretical Contribution: The paper introduces a unified theoretical framework that extends established single-domain scaling law theories—such as the Quantization Model and Linear Regression Model—to multi-domain settings. It provides a mechanistic understanding of scaling laws, moving beyond purely empirical curve-fitting.

2.Intuitive and Insightful Structural Assumption: The proposed “Shared Head, Disjoint Tail” assumption effectively captures the intuition that domains share foundational skills but differ in specialized knowledge. This offers a clear conceptual foundation for modeling multi-domain interactions and plays a central role in the theoretical derivations.

3.Well-Reasoned and Generalizable Mechanism Explanations: Two key mechanisms—Capacity Competition and Noise Reduction/Driven Shift—are introduced to explain inter-domain loss coupling and optimal mixture shifts. These mechanistic insights offer greater interpretability and generalizability compared to black-box empirical formulas from prior work.

4.Rigorous Mathematical Modeling: The problem is formally framed using convex optimization (for loss prediction) and bi-level optimization (for mixture search). The derivations, including approximate closed-form solutions and gradient characterizations, are logically sound and enhance the theoretical credibility.

5.Comprehensive and Convincing Experimental Validation: Experiments on models of 200M and 700M parameters confirm the theoretical predictions, demonstrating both superior fitting accuracy over empirical baselines and the ability to predict optimal training mixtures. The strong alignment between theory and experiment reinforces the paper’s claims.

6.High Parameter Efficiency and Practical Relevance: Compared to existing empirical approaches, the proposed model achieves better performance using significantly fewer free parameters. This efficiency is particularly valuable for scalability and practical deployment in large-scale model training.

7.Clear Advance Over Existing Work: By providing theoretical explanations and mechanistic insights, in contrast to purely empirical methods such as those in Shukor (2025) and Ye (2024), the paper clearly establishes its novelty and contributes meaningfully to the literature.

Weaknesses:

1.Idealized Assumptions and Their Validation: The core "Shared Head, Disjoint Tail" assumption is necessarily an approximation, as tails in practice are not fully disjoint. While Section 3.1 argues that overlap leads to an approximately constant loss, this relies on a simplified skill-learning model. To strengthen this claim, it would be valuable to empirically quantify the degree of tail overlap (e.g., among Code, Math, and Text domains) in the experiments.

2.Limited Experimental Scope: The experiments mainly focus on medium-scale models (200M, 700M), lacking validation on larger-scale models or more diverse tasks.

---

> ### Author Rebuttal · Authors · 2026-03-31
>
> We sincerely thank the reviewer for valuable feedback and constructive comments.
>
> - Anonymous link: https://anonymous.4open.science/r/ICML2026_response-5FEB/Review1.pdf
>
> ## Responses
>
> > Weakness 2.Limited Experimental Scope
>
> - **New Experiments with Post-Submission Data Release:** Our initial scope was constrained by computational resources and data availability. Recently, Shukor et al. kindly released their experimental data for the 7-domain SlimPajama mixtures (after the ICML submission deadline). We incorporate this new data into our evaluation.
>     - **Fitting Accuracy**: We replicated the experiments from Section 4.2 using the 7-domain data and the original additive law fitting code from Shukor et al. (2025), the current SOTA. As shown in the table below, our theoretical law from Section 3.3 consistently achieves a lower MRE across different model sizes and data sizes. Results are listed in Table 1 in the link.
>     - **Optimal Mixture Extrapolation:** Furthermore, we used our Section 3.3 model—fitted solely on their 310M parameter / 20B token mixtures—to predict the optimal data mixture for a 1B parameter model trained on 25B tokens (by scaling $N$ and $D$ proportionally). We then trained 1B models using both our predicted mixture and the optimal mixture reported by Shukor et al. (Table 15). Our predicted mixture strictly outperforms their optimal mixture, achieving a superior loss of **2.059** compared to their **2.063**.
>
> - **Complementary Results for Original Experiments**
> To address the reviewer’s concerns, we have also strengthened our original experiments:
>     1. Extended Comparison (Section 4.2): We computed and compared the optimal mixtures for the 17-domain Pile dataset. For a 1B model trained on 25B tokens, our mixture outperforms the one derived from the Shukor et al. additive law. Detailed results are available in Figure 1 in the link.
>     2. Hyperparameter Correction (Section 4.3): We corrected a suboptimal hyperparameter selection in our initial Section 4.3 experiments. With the corrected optimal hyperparameters, our law outperforms the baselines more significantly than previously reported. Please see Table 3 and Figure 2 in the link.
>
> >Q1: Under the “Shared Head, Disjoint Tail” assumption, how can the degree of tail overlap be quantified?... and how does it affect the final ‘effective capacity requirement’ function?
>
> - Within our simplified skill-learning model, the status of overlapping skills (learned vs unlearned) will likely be more stable as the mixture changes, which means they are more likely to take up a constant fraction of the "effective capacity". Therefore, a reasonable approximation of the tail overlap effect is to subtract a constant from the total effective capacity, which is basically the same as our current model.
> - To empirically validate this approximation, we conducted a synthetic experiment simulating $K=7$ domains with a 40% global tail overlap. When fitting our disjoint-tail model to this highly overlapping data, the MRE was remarkably low at just 0.07%. This confirms that even under violations of the disjoint assumption, our model remains a good approximation.
> - Furthermore, our experiments on the Pile dataset naturally contain domains with high tail overlap, such as GitHub and StackExchange, and the more broadly overlapping Pile-CC. Despite this overlap, our model achieves the lowest MRE.
>
> >Q2: In the bilevel optimization problem, you mention that the optimal mixture is not equal to the target distribution. Could you provide more intuitive examples to illustrate the magnitude and pattern of this deviation?
>
> - See Figure 3 in the link.
>
> >Q3: How applicable is your method to larger scales.. Are there any preliminary experiments or theoretical extrapolations?
>
> - Theoretical Extrapolation: While our current theoretical framework primarily analyzes fixed model and dataset sizes, it establishes a principled foundation for predicting optimal data mixtures at larger, unseen scales. Scaling our framework requires solving two specific modeling challenges: (1) establishing a precise analytical mapping from raw parameter counts to effective model capacity, and (2) characterizing how the data-dependent noise term scales with varying dataset sizes.
>
> - Preliminary Experiments: We have conducted preliminary experiments that demonstrate our method's extrapolation capabilities. In our Section 4.3 experiments, we first fitted our model at the 200M parameter scale (setting the effective model capacity to 200). Using these fitted parameters, we then computed the optimal data mixture for a 700M parameter model by scaling the effective capacity proportionally to 700. We found that the extrapolated 700M mixture strictly outperforms the optimal mixture derived for the 200M model (as well as the baseline). This indicates that our model correctly predicts the shift of the optimal mixture as model size increases. Please find the details in Table 3 and Figure 2 in the link.

---

> > ### Author Rebuttal · Reviewer_6QL9 · 2026-04-06
> >
> > Thank you for your comprehensive and thoughtful rebuttal. I have reviewed your detailed responses and the newly added experimental results.
> >
> > The additional experiments are compelling and effectively address the core concerns regarding experimental scope and assumptions. The work done post-submission significantly strengthens the paper’s contributions and empirical support.
> >
> > I appreciate the effort and clarity in your revision. While I must acknowledge that the theoretical formulation in this paper is quite dense and falls somewhat outside my immediate expertise.
> >
> > This is solid work, and I wish you the best of luck with the final decision.

---

> > > ### Author Response · Authors · 2026-04-07
> > >
> > > Thank you for your encouraging feedback and for acknowledging the effort behind our rebuttal. We were very fortunate that the new experimental data became publicly available shortly after the submission deadline, which allowed us to apply our framework to it. We are very glad that these new results successfully address your concerns and solidify the paper's empirical contributions. Finally, thank you once again for the time and effort you have dedicated to reviewing our paper and carefully reading our rebuttal responses.

---

### Decision · Program_Chairs · 2026-04-30

**Decision:**

Accept (regular)

**Comment:**

This paper discusses how to find ideal mixture weights for weighing data from different domains in order to optimize performance for a given target domain. The reviewers were in consensus that the paper is mathematically sound and has interesting empirical experiments.

The main concern raised by the reviewers revolved around the so-called “shared head, disjoint tail” assumption in the paper where different domains have a shared set of features/skills and differ in specialized data. This assumption was validated by the authors in new experiments in the rebuttal period.

Reviewer j8aK had concerns regarding the clarity of the exposition and the various approximations used in the derivations. The authors have provided reasonable justifications for some of their choices and have said that they will modify make the notation more precise. Reviewer n3Zm had similar concerns, about the assumption that the different domains have a single shared teacher.

The other comments and concerns raised by the reviewers have been largely addressed by the rebuttal provided by the authors. The theoretical aspects of the paper could be strengthened significantly. But the empirical results in the paper and the new ones added during the review period are interesting. I would suggest that the authors carefully rewrite the mathematical parts of the paper make the arguments more precise (e.g., it was quite unclear to me what even skills are in the problem setup, or what “frequency” means on Line 209).